

# 1 Non-mycorrhizal root-associated fungi increase soil C stocks and
# 2 stability via diverse mechanisms

Emiko K. Stuart[1*], Laura Castañeda-Gómez[1,2], Wolfram Buss[3], Jeff R. Powell[1], Yolima Carrillo[1]
[1]Hawkesbury Institute for the Environment, Western Sydney University, Richmond, NSW 2753, Australia
[2]SouthPole - Environmental Services, Technoparkstrasse 1, Zürich 8005, Switzerland (Present address)
[3]Research School of Biology, Australian National University, ACT 2601, Australia
*Correspondence to*: Emiko K. Stuart (e.stuart@westernsydney.edu.au)
**Abstract.** While various root-associated fungi could facilitate soil carbon (C) storage and therefore aid climate change
mitigation, so far research in this area has largely focused on mycorrhizal fungi, and potential impacts and mechanisms for
other fungi are largely unknown. Here, we assessed the soil C storage potential of 12 root-associated, non-mycorrhizal fungal
isolates (spanning nine genera and selected from a wide pool based on traits potentially linked to soil C accrual) and
investigated fungal, plant and microbial mediators. We grew wheat plants inoculated with individual isolates in chambers
allowing continuous $^{13}$C labelling. After harvest, we quantified C persistence, and pools of different origin (plant vs soil) and
of different stability with long-term soil incubations and size/density fractionation. We assessed plant and microbial
community responses, as well as fungal physiological and morphological traits in a parallel *in vitro* study. While inoculation
with three of the 12 isolates resulted in significant total soil C increases, soil C stability improved under inoculation with most
isolates – as a result of increases in resistant C pools and decreases in labile pools and respired C. Further, these increases in
soil C stability were positively associated with various fungal traits and plant growth responses, including greater fungal hyphal
density and plant biomass, indicating multiple direct and indirect mechanisms for fungal impacts on soil C storage. We found
more evidence for metabolic inhibition of microbial decomposition than for physical limitation under the fungal treatments.
Our study provides the first direct experimental evidence in plant-soil systems that inoculation with specific non-mycorrhizal
fungal strains can improve soil C storage, primarily by stabilising existing C. By identifying specific fungi and traits that hold
promise for enhancing soil C storage, our study highlights the potential of non-mycorrhizal fungi in C sequestration and the
need to study the mechanisms underpinning it.

## 25 1 Introduction

Despite soils having the capacity to sequester large amounts of atmospheric $CO_2$ and mitigate catastrophic climate change, the
full potential of soil carbon (C) sequestration is yet to be realised (Field and Raupach, 2004; Scharlemann et al., 2014;
Schlesinger, 1990). Moreover, rather than being protected, soils are becoming increasingly degraded globally due to intensive
agricultural practices - a situation that may worsen as C loss potentially accelerates with future climate scenarios (Hannula and
Morriën, 2022; Lal, 2018). While soil C sequestration is becoming more broadly recognised as an important climate mitigation





strategy, and as an approach to recover the multiple ecosystem services provided by soil C (Kopittke et al., 2022), its successful
implementation first requires understanding of processes underpinning soil C storage (Dynarski et al., 2020; Smith and Wan,
2019; Von Unger and Emmer, 2018). Knowledge of soil C storage has improved substantially in recent years, with it now
understood to result from the balance of multiple, dynamic processes (that are further complicated by pedoclimatic context)
determining C inputs to soil and their stabilisation - which ultimately determine the persistence of soil C at the ecosystem scale
(Cotrufo and Lavallee, 2022; Derrien et al., 2023; Dignac et al., 2005; Dynarski et al., 2020; Jackson et al., 2017; Schmidt et
al., 2011). Soil microbes act as key participants of these processes: they regulate the persistence of soil C primarily via their
abilities to mineralise soil organic matter, which determine how long C of plant or microbial origin persists in soil, and can
also influence how much C is available for stabilisation from their necromass and from plant inputs. However, the soil
microbial community is complex, and largely unknown; hence, referred to as a "black box" (Mishra et al., 2023; Tiedje et al.,
1999). Within this black box, fungi, both free-living and plant-associated, are considered particularly important for soil C
storage; however, their impacts on soil C storage are both multifaceted and diverse.
The complexity in fungal impacts on soil C storage firstly arises from their abilities to influence both soil C inputs and
stabilisation via multiple direct and indirect mechanisms occurring simultaneously (Hannula and Morriën, 2022; Kallenbach
et al., 2016; Liang et al., 2019; Starke et al., 2021). In general, fungi that are present in soil (1) all produce hyphae and with
them hyphal C inputs, (2) can alter plant health, growth, and C chemistry and allocation to soil, and (3) can influence the rest
of soil microbial community structure and composition, thus impacting fungal-, plant-, and microbial-derived C, respectively
(Clocchiatti et al., 2020; Hannula and Morriën, 2022; Rai and Agarkar, 2016; Stuart et al., 2022). All of these inputs, but
particularly fungal and plant C, are potentially available for soil C storage but they require stabilisation in order to persist in
soil long term. The broad and efficient enzymatic capabilities and extensive mycelial structure of fungi, as compared to the
rest of the microbial community, allow them to competitively obtain soil C and transform it so that it can be readily sorbed
and stabilised onto mineral surfaces (Boer et al., 2005; Hannula and Morriën, 2022). In addition, fungal necromass is
considered to have a particularly strong affinity for mineral surfaces and is therefore an important source of stabilisable C
(Sokol et al., 2019). The impact of fungi on soil structure and spatial heterogeneity, including promoting aggregate formation
by enmeshing soil particles with their hyphae and producing various extracellular biopolymers, further protects C by physically
constraining microbial decomposition, leading to greater persistence (Berg and Mcclaugherty, 2014; Dynarski et al., 2020;
Kleber et al., 2011; Lehmann et al., 2017; Lützow et al., 2006; Schmidt et al., 2011).
These various impacts of fungi on soil C storage are further complicated by fungal diversity, which occurs at the inter-genus,
inter-species, and even down to the sub-species level (Andrade et al., 2016; Hiscox et al., 2015; Johnson et al., 2012; Juan-
Ovejero et al., 2020; Plett et al., 2021). In plant-soil ecosystems, fungi exist either as free-living saprotrophs or as plant-
associated fungi, including mycorrhizal, endophytic, and parasitic fungi (Rai and Agarkar, 2016). Saprotrophic fungi are often
assumed to promote soil C output, as they decompose soil organic matter due to being outcompeted by mycorrhizal fungi for



plant C exudates, but as decomposition can increase the availability of C to be sorbed onto mineral surfaces, thereby fostering
soil C stability, their net impacts on soil C storage may need further exploration (Frąc et al., 2018; Hannula and Morriën, 2022;
Lehmann and Rillig, 2015). Meanwhile, much of the research on the impacts of plant-associated fungi on soil C has focused
on mycorrhizal fungi, particularly arbuscular mycorrhizal fungi and ectomycorrhizal fungi due to their dominance in their
respective habitats (Jackson et al., 2017; Smith and Read, 2008). These fungi have additional impacts, to the general fungal
impacts outlined above, on the inputs, stabilisation, and persistence of C. As they transform and funnel plant C belowground,
mycorrhizal fungi can increase and modify the quality of C inputs, for example by synthesising melanin for cell walls, which
is considered to be highly stable and has been associated with decreased hyphal decomposability and increased soil C content
(Fernandez and Kennedy, 2018; Fernandez and Koide, 2013; Zak et al., 2019; Zhu and Michael Miller, 2003). Due to their
nutrient requirements and abilities to mine soil resources, they are thought to be strong competitors against saprotrophs for not
only plant C but also soil nutrients, thereby suppressing microbial respiration, and resulting in greater C persistence (Gadgil
and Gadgil, 1971; Averill and Hawkes, 2016). Some mycorrhizal fungi have limited abilities to directly and partially decay
organic matter, and they can also prime saprotrophic microbes to decompose pre-existing soil C, thus having the potential to
decrease C persistence – though their net impact on soil C storage is not well understood (Frey, 2019). Despite the large
diversity amongst fungi in plant-soil ecosystems, influences of non-mycorrhizal fungi, particularly other plant-associated
fungi, on soil C storage have not been studied in as great detail compared to mycorrhizal fungi but do hold promise. For
example, endophytic fungi could potentially be important for soil C storage due to their abilities to produce melanin and
promote plant growth (Berthelot et al., 2017; He et al., 2019; Mandyam and Jumpponen, 2005; Rai and Agarkar, 2016).
However, there are conflicting reports regarding their lifestyles, benefits or harms imposed on host plants, enzymatic and
nutrient acquisition ability, or even whether they produce extraradical mycelium, suggesting there may be wide functional
variation or plasticity within this fungal group (Addy et al., 2005; Mukasa Mugerwa and Mcgee, 2017; Rai and Agarkar, 2016).
To better understand the diversity of fungal impacts on soil C storage, particularly soil C stability, more focus is needed on
fungal types other than mycorrhizal fungi.
There is growing interest in searching and screening for organisms that, in addition to supporting plant productivity, may
improve soil C storage in agricultural systems (Kaminsky et al., 2019; Islam et al., 2021; Salomon et al., 2022). Thus far,
mycorrhizal fungi have received much attention in this area due to their well-established impacts on plant health and soil C.
However, as discussed above, other fungal types may also offer advantages to soil C storage and plant productivity but have
been largely unexplored. With this objective in mind, in the current study we aimed to determine the potential of diverse non-
mycorrhizal fungi to impact soil C stocks, formation (by impacting the origin of soil C), and persistence (by impacting C pools,
dynamics, and fractions), and to investigate the mechanisms underpinning these impacts, both direct and indirect. We assessed
12 separate fungal species (spanning nine genera in the orders Chaetosphaeriales, Helotiales, and Pleosporales), isolated from
roots collected from multiple soil environments across Australia and screened for traits that may support plant growth and soil
C storage, such as capabilities to capture and solubilise nutrients from the soil. In a pot study, we inoculated spring wheat



(*Triticum aestivum*), an important cereal crop, with one of the 12 fungi and grew the plants for a full life cycle in [13]C-depleted
$CO_2$ growth chambers to homogeneously label the plants during the full growth cycle, in order to distinguish soil C from plant-
derived soil C. Following harvest, we assessed total C and its isotopic composition, and assessed C distribution among pools
of different stability and persistence (labile, intermediate, and resistant) via four-month soil incubations, and evaluated the
contribution of soil and plant C to these pools using isotopic analysis. These incubation-based assessments were accompanied
by size and density fractionation analyses to quantify mineral-associated organic matter (MAOM), aggregate carbon (AggC),
and particulate organic matter (POM). We then measured traits of the fungi and of the plants and microbial community to
explore the potential direct and indirect mechanisms behind these impacts, respectively. We hypothesised that if a fungal
species increased total soil C storage, this would be due primarily to increasing plant C inputs by supporting plant growth and
also to stabilising existing soil C - so that fungi-driven increases in total soil C would be associated with more persistent and
stable pools and fractions of C. We expected that these changes to soil C would be associated with fungal traits, alluding to
direct mechanisms, as well as to increases in plant growth and shifts in microbial community composition, alluding to indirect
mechanisms.

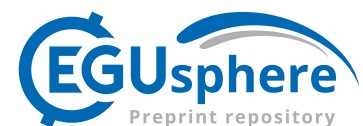

## 2 Materials and methods

The overall study design consisted of a wheat growth pot experiment, in which changes to soil, plant, and soil microbial communities in response to fungal inoculation were assessed, and a separate *in vitro* fungal growth assay, to measure fungal traits that could potentially be linked to observations made in the main experiment (Fig. A1).

### 2.1 Experiment set up and maintenance

Twelve fungal isolates were originally obtained from plant roots and screened for traits that may support plant growth and soil C storage by Loam Bio Pty Ltd (Orange, New South Wales, Australia). The fungal isolates, including endophytic fungi and potentially saprotrophic or other fungi, comprised: a *Thozetella* species, a *Paraconiothyrium* species, three *Darksidea* species, a *Leptodontidium* species, a *Clohesyomyces* species, two *Phialocephala* species, an *Acrocalymma* species, a *Periconia* species, and an *Ophiosphaerella* species. Pure cultures of these isolates were maintained on 1/10 strength potato dextrose agar (PDA). Surface-sterilised (2% NaOCl) and moistened seeds of Australian wheat cultivar Condo (*Triticum aestivum*) were incubated at room temperature for 48 h. Soil was obtained from an agricultural field, sieved through 2 mm, and was a clay loam (4.3% C, 0.39% N, pH 5.85; Table B1).

The experimental setup consisted of seven planted replicates inoculated with one of the 12 fungal isolates, and six replicates of uninoculated planted pots, distributed among six $CO_2$-controlled growth chambers (Climatron-1260; Thermoline, Wetherill Park, New South Wales, Australia) that had been modified to achieve continuous $^{13}$C-labeling of plant tissues. For "planted" replicates, three 7 mm agar squares from actively growing 1/10 PDA fungal culture plates were placed near three sterile seeds in 2 L plastic pots (at a depth of 2-3 cm) containing 1800 g of non-sterile soil. Uninoculated planted pots ("absent/control") received three agar squares from uninoculated plates. Each agar square contained approximately 1.3 mg C. Smaller pots (containing 500 g of soil) for "unplanted" control pots (see below) were set up three days later using two agar squares, as controls for impacts of fungi in the absence of plants, adding to 142 pots in total. After 10 days of growth, seedlings were thinned to one per pot.

Pots were regularly and uniformly watered with tap water. Pots within each tub and tubs within each chamber were randomly relocated four times throughout the experiment. The chamber atmosphere was sampled weekly to confirm that the atmospheric $CO_2$ was sufficiently depleted in $^{13}$C via a pump system into a Tedlar® SCV Gas Sampling Bag and δ$^{13}$C analysis in a PICARRO G2201i isotopic $CO_2$/$CH_4$ analyser (Picarro Inc., Santa Clara, CA, USA).

### 2.2 Harvest and plant biomass measurement

Once the plants had senesced and the grain had ripened, at 18th weeks of growth, wheat spikes and shoots were cut off, dried at 70°C and weighed. The intact root-containing soil was preserved in the pots by freezing at -20°C immediately after shoots



were cut to stop all decomposer activity to retain the C status generated by the treatment until ready for subsampling and processing. After two days of thawing at 4°C, soil was removed from the pots and a subsample for fractionation analysis was collected from near the root crown and oven-dried at 40°C. The main root system was gently shaken of soil and 1/3 of the roots were cut, washed, patted dry, frozen at -20°C prior to root morphology measurement. The rest of the soil was homogenised before subsamples collection. A subsample for phospholipid fatty acid (PLFA) analysis was frozen at -20°C. A subsample for soil moisture content was weighed and dried at 105°C. A subsample for soil incubations was oven-dried at 40°C and sieved at 2 mm, and of this, a further subsample for isotope analysis was dried at 105°C. To obtain total root mass, first the root/soil ratio outside the main root system was estimated by collecting the root mass of the remaining soil (after all subsampling) via wet sieving (500 µm) and oven-drying at 40°C. The root mass of the soil subsamples was calculated using this ratio and the amount of soil in all subsamples.

**2.3 Root morphology**

To evaluate root morphology, a potential indirect mechanism for fungal impacts on soil C storage, washed, dried, and frozen root subsamples were arranged with minimal overlap for digital scanning (Epson Expression 11000XL scanner, Epson, Macquarie Park, Australia). Images were analysed with WinRhizo Pro software 2015 (Regent Instruments Inc., Quebec City, Canada) to obtain root average diameter (mm), specific length as the ratio of length to dry mass (cm mg$^{-1}$), tissue density as mass per unit volume (g cm$^{-3}$), specific surface area as the ratio of area to dry mass (cm$^2$ g$^{-1}$), and branching as the number of forks per unit of mass (number mg$^{-1}$). Following root morphology assessment, the root subsample was oven-dried at 40°C for determination of total root mass.

**2.4 Plant and soil isotope and chemical analysis**

To determine the contribution of soil- versus plant-derived C to total C in soils under wheat, isotopic compositions and C/N content of ground shoots and soil were assessed using an elemental analyser interfaced to a continuous flow isotope ratio mass spectrometer (UC Davis Stable Isotope Facility, Davis, California, USA). The proportion of original soil C present in the soil of each pot after plant growth was calculated via isotopic partitioning following Eq. (1):

$$\text{Soil proportion.Soil} = \frac{(\delta 13C_{Soil} - \delta 13C_{UP-Soil})}{\delta 13C_P - \delta 13C_{UP-Soil}},$$

where $\delta^{13}C_{Soil}$ is the $^{13}C$ isotopic composition of soil measured in each planted pot, $\delta^{13}C_{UP-Soil}$ is the mean $^{13}C$ isotopic composition of soil in unplanted controls, and $\delta^{13}C_P$ is the $^{13}C$ isotopic composition of the plant shoots in each planted pot. The plant C proportion (including C from other biological sources) was defined as 1 minus the soil C proportion. These proportions were then applied to the measured C concentrations in each pot to calculate plant- and soil-derived C amounts.



## 2.5 Soil incubations

To evaluate the impact of fungal isolates on overall C persistence and C distribution across pools of different stability (labile, intermediate, and resistant), we assessed microbial $CO_2$ production during 135-day laboratory incubations of soil harvested at the time of wheat harvest under standard temperature and moisture conditions, and fitted a decay model to estimate decay kinetic parameters. Kinetic parameters derived from mid- to long-term soil incubation are sensitive functional measures of changes in the distribution and stability of C pools resulting from previous exposure to experimental treatments (Carney et al., 2007; Carrillo et al., 2011; Jian et al., 2020; Langley et al., 2009; Taneva and Gonzalez-Meler, 2008). Measured $CO_2$ production rates over time were fitted to a two-pool exponential decay model to estimate the size of the labile and intermediate C pools and their mean residence time (MRT; Cheng and Dijkstra, 2007; Wedin and Pastor, 1993). The size of the resistant pool was calculated as the difference between the total measured organic C and the sum of the estimated labile and intermediate pools. This same procedure was also applied to the portion of $CO_2$ that was released from the originally present soil C (soil-derived C, i.e. not plant-derived C), which was determined via isotopic partitioning of plant vs. soil-derived $CO_2$. Based on these, we calculated total $CO_2$ released from plant- and soil-derived C during the full length of the incubation. See Supplementary Methods for full details on incubations, isotopic partitioning, and decay modelling.

## 2.6 Soil fractionation analysis

Soil fractionation analysis was performed as an alternative method to soil incubations for understanding fungal impacts on C stability and potential persistence. Hereafter we refer to the pools measured via fractionation analysis as "fractions", as opposed to "pools" measured via soil incubations. The analysis was performed according to a method developed by (Poeplau et al., 2017; Poeplau et al., 2018) and adapted by Buss et al. (2023, in review) involving high throughput physical fractionation into conceptually designed soil C fractions - mineral-associated organic matter (MAOM), aggregate carbon (AggC), and particulate organic matter (POM). See Supplementary Methods for further details.

## 2.7 Soil PLFA analysis

Total microbial community size and composition are also potential indirect drivers of fungal impacts on soil C storage. Microbial PLFAs in soils were extracted from 2 g of freeze-dried soil harvested from the wheat growth experiment, following the high throughput method developed and described by Buyer and Sasser (2012; see Supplementary Methods).

## 2.8 *In vitro* fungal assessment

To assess morphological and chemical properties of the fungal isolates (used in the wheat growth experiment) as potential drivers of fungal impacts on soil C storage, a separate *in vitro* plate assay was performed using 1/2 PDA plates incubated in the dark at 23-25°C (see Supplementary Methods). Radial growth rate was calculated by measuring colony areas every two-



to-three days using ImageJ (National Institutes of Health, Bethesda, Maryland, US; Schneider et al., 2012). Growth rate was
calculated by subtracting the colony area from an earlier sampling point from that of the following sampling point. Hyphal
density was calculated as the final fungal biomass per final colony area. C and N content were measured by Dumas combustion
using a El Vario cube analyser (Elementar, Langenselbold, Germany).

**2.9 Data and statistical analysis**

ANOVA of soil C properties and experimental variables was performed in R (v. 4.1.2; R Core Team, 2021), followed by
Dunnett's post-hoc test to determine which treatment groups were significantly different to the uninoculated control group or
Tukey's post-hoc test to determine significant differences between inoculated groups. Principal component analysis (PCA)
and redundancy analysis (RDA) were performed using the vegan package in R (Oksanen et al., 2020). Missing values in the
PCA and RDA datasets were replaced with the variable mean.
Curve fitting of $CO_2$ rate dynamics was done using the non-linear modelling platform in JMP 16.1.0 and the biexponential
four-parameter decay model using all replicates of a treatment. We used nonlinear least square curve fitting to test if the models
were significantly different between a fungal treatment and uninoculated control, using the nls function in R.





## 3 Results

### 3.1 Several non-mycorrhizal fungal species increased soil C under wheat plants

We inoculated wheat plants (*Triticum aestivum*) with one of 12 fungi (non-mycorrhizal) isolated from plant roots. After four months of plant growth, there was a positive but varied effect of fungal inoculation on soil C content compared to the uninoculated control group ($p < 0.05$; Fig. 1, Table B2). This effect was not observed in soils that received the same fungi but were unplanted ($p = 0.22$; Fig. 1). We found significant isolate-specific increases in soil C content of the planted treatments under inoculation with *Thozetella* sp., *Darksidea* sp. 3, and *Acrocalymma* sp., relative to the uninoculated control, of 9.4% (percentage of change), 7.5, and 7.8, respectively. Nitrogen levels were generally higher in the soils of the inoculated and planted treatments compared to the uninoculated control and were generally higher in the treatments where C was also higher (Table B2).

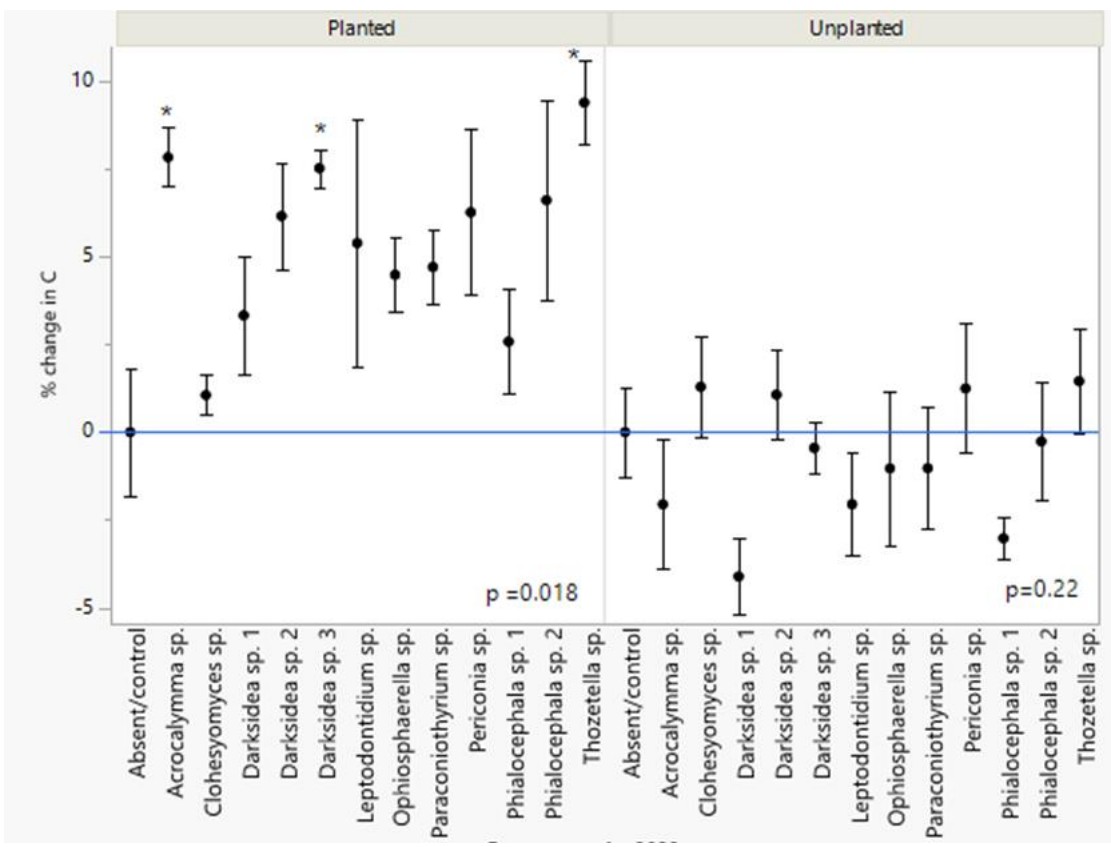

**Figure 1. Changes in total soil C under inoculation with different fungal isolates compared to an uninoculated control. Values indicate percentage of change relative to mean of uninoculated control (blue line). Error bars indicate standard error, n=7 for inoculated treatments, n=6 for control. ANOVA results for planted and unplanted are presented.**



**Asterisks indicate significant differences with control (Dunnett test, p < 0.05). C concentrations are presented in Table**
**B2.**
**3.2 Fungi-dependent increases in soil C are associated with changes in soil C pools, origin, and persistence**
To understand the underlying mechanisms of the fungal isolate-dependent increases in soil C content and potential shifts in
sources and stability of the resulting soil C, we performed C isotope analysis, soil incubations, and soil C fractionation analysis.
Isotopic partitioning of C into plant- and soil-derived C revealed how changes in these pools contributed to changes in total
soil C (Fig. 2a, Table B2). Planting reduced total soil C, compared to initial C prior to planting, as expected due to C inputs
stimulating decomposition (rhizosphere priming). This reduction was due to decreases in soil-derived C, which were generally
not counteracted by newly added plant-derived soil C - which on average represented 3.8% (±0.2) of total soil C. Some
increases in total soil C compared to the planted uninoculated controls could be explained by plant- and soil-derived C. Namely,
one of the fungal treatments whereby total soil C significantly increased (*Thozetella* sp.) exhibited higher amounts of plant-
derived C - at a level that was marginal in its non-significance. However, overall, the higher total soil C content relative to
controls corresponded more closely with higher soil-derived C (R = 0.93, p < 0.01), than with plant-derived C (R = 0.02, p =
0.83). All three fungal treatments resulting in significant increases in total soil C showed increases in soil-derived C but these
were not statistically significant.





**Figure 2. Distribution of total soil C in plant- and soil-derived pools (A) and among labile, intermediate, and resistant pools (B) in soil under inoculation with different fungal isolates or under no inoculation (Absent/control). (A): Plant- and soil-derived C from C isotope partitioning (see Materials and methods). Black asterisks indicate significant differences in total C with control and white asterisks differences in plant-derived soil C with control (Dunnett test, p < 0.1); (B): Pools estimated from decay models derived from soil incubation (see Materials and methods). Crosses**



**indicate significant differences in the dynamics of total C decomposition (decay curves models, Table B3) compared to**
**the uninoculated control. Asterisks indicate significant differences in total C or resistant C against control (Dunnett**
**test, p < 0.05). Error bars indicate standard error of total C, n=7 for inoculated treatments, n=6 for uninoculated**
**control. Note y axis scale.**
Incubation of soils after plant harvest demonstrated impacts of several fungal species on the dynamics of C decomposition and
the distribution of C among soil pools of different stability. The dynamics of total C decomposition (decay curves models
derived from incubations) were significantly different to the control under half of the isolates (Table B3, Fig. A2). These
include the three isolates that produced higher total C pools: *Thozetella* sp., *Darksidea* sp. 3, and *Acrocalymma* sp. Soil-derived
C decomposition curves (from isotopic partitioning of respiration) were also significantly different to the controls under the
same fungal treatments as well as *Leptodontidium* sp. Estimated pools from these decay curves showed significantly higher
total resistant C (up to 86% of C), compared to controls (76% of C), under eight of the 12 isolates, including the three treatments
where total C increased the most (Fig. 2b, Table B3). In terms of other pools, MRT of the total labile C was significantly lower
under inoculation with *Darksidea* sp. 1 compared to the control, whereas MRT of the soil-derived labile C was significantly
higher under inoculation with *Periconia* sp. (Table B3). In terms of intermediate pool MRTs, controls and fungal treatments
were not significantly different.
Soil incubations and partitioning of respiration revealed fungal effects on the degree of persistence of total C, soil-derived C,
and plant-derived C over time, which we assessed as the proportion of what was present at harvest that was respired over the
full incubation. Significantly lower proportions of total and soil-derived C were respired under all fungal treatments compared
to the controls (p < 0.001; Fig. A3), indicating increased persistence. In contrast, plant-derived respired C was significantly
lower (more persistent) than the controls only with *Thozetella* sp. (p < 0.05).
From fractionation analysis, %C and %N of the AggC fraction, i.e. the fraction of intermediate stability whereby C is protected
in aggregates, were found to have significant fungal effects, with *Thozetella* sp. and *Periconia* sp. exhibiting significantly
higher levels of both C and N, and *Ophiosphaerella* sp. and *Phialocephala* sp. 1 exhibiting significantly higher levels of N
compared to controls (Table B4). Significant fungal effects were not observed in the MAOM and POM fractions.
We performed PCA to identify soil C properties associated with fungi-driven increases in soil C (Fig. 3). Most of the variance
was explained by PC1 and 2 (58%). Greater total soil C (C) was closely associated with soil-derived C (SC), but not plant-
derived C (PC), at time of harvest and soil N. Soil C was also related with the resistant C pools (total (TRC) and soil-derived
(SRC)). The treatments with lowest total soil C (mainly the control, followed by *Clohesyomyces* sp., and *Phialocephala* sp. 1;
Fig. 1) were associated with higher proportions of total and soil-derived C respired during incubation indicating that the C
remaining at harvest was inherently less persistent. %C of the AggC and MAOM fractions, considered to be more stable
fractions of C, were not clearly associated with soil C or the resistant C pools, nor with any fungal treatments.



**Figure 3. Fungi-dependent increases in soil C largely relate to measures for soil C stability. Principal component analysis showing soil C properties (red text) associated with various fungal isolates (symbols). Soil C properties were measured via isotope analysis, soil incubations, and fractionation analysis of soil from wheat experiment. Soil C property abbreviations: AFC, aggregate C fraction %C; C, %C; MFC, MAOM fraction %C; N, %N; PC, plant-derived C (µg g⁻¹ soil); PFC, POM fraction – %C; PRpP, plant-derived C respired proportion; SC, soil-derived C (µg g⁻¹ soil); SIC, soil-derived intermediate C (µg C g⁻¹ soil); SLC, soil-derived labile C (µg C g⁻¹ soil); SRC, soil-derived**





**resistant C (µg C g$^{-1}$ soil); SRpP, soil-derived C respired proportion; TIC, total intermediate C (µg g$^{-1}$ soil); TLC, total**
**labile C (µg g$^{-1}$ soil); TRC, total resistant C (µg g$^{-1}$ soil); TRpP, total C respired proportion.**
**3.3 Fungi-dependent increases in soil C and its stability and persistence are positively associated with plant growth and**
**microbial community composition**
We assessed plant and microbial community variables, including plant biomass, shoot C/N content, root morphology, and total
microbial community size and composition derived from PLFA analysis. Overall, while variation among fungal isolates was
observed, no significant differences were observed between the inoculated and uninoculated plants for any of the plant or
microbial community variables measured, although average spike mass of *Thozetella*-inoculated plants was significantly
higher than that of uninoculated plants (Table B5-6).
To identify plant and microbial community variables potentially involved in the fungal isolate-dependent changes in soil C
properties, we performed RDA using plant and microbial community data and the soil C property data used in the PCA (Fig.
4). Variance explained by RDA1 and 2 was 14.28 and 4.72%, respectively. The cluster of soil C properties that were found to
be closely associated with *Thozetella* sp. in the PCA (e.g. soil-derived C, resistant C pools; Fig. 3) also trended positively with
plant biomass and growth (spike and shoot mass, shoot C/N ratio, and root fork number) and with the PLFA-assessed fungal
to bacterial ratio. *Acrocalymma* sp. and *Darksidea* sp. 3 were more associated with root growth traits, and were also associated
with plant-derived C. The low soil C treatments (uninoculated control, *Clohesyomyces* sp., and *Phialocephala* sp. 1) and their
associated soil C properties (i.e. respired C) were related to shoot C and N.






**Figure 4. Fungal treatments resulting in increased soil C and its stability are associated with plant growth. Redundancy analysis showing microbial community and plant variables (blue text) driving changes in soil C properties (red text) associated with various fungal isolates (symbols). Soil C properties were measured via isotope analysis, soil incubations, and fractionation analysis of soil from wheat experiment. Microbial community and plant variables were measured using samples harvested from the wheat experiment. Microbial community (M.) and plant (P.) variable abbreviations: M.AB, actinobacteria (% of total community); M.AMF, arbuscular mycorrhizal fungi (% of total community); M.F, fungi (% of total community); M.FB, fungal to bacterial biomass ratio; M.GNB, gram negative bacteria (% of total**





**community); M.GPB, gram positive bacteria (% of total community); M.TC, total community size (μg PLFA g$^{-1}$ soil);**
**P.RADi, root average diameter (mm); P.RF, root fork number (g$^{-1}$); P.RLDe, root length density (cm g$^{-1}$); P.RLV, root**
**length per volume (cm m$^{-3}$); P.RM, root mass (g); P.RS, root/shoot ratio; P.RSA, root specific surface area (cm$^2$ g$^{-1}$);**
**P.RSDe, root specific density (g cm$^{-3}$); P.S15N, shoot δ15N (‰); P.SC, shoot %C; P.SCN, shoot C/N ratio; P.SM, shoot**
**mass (g); P.SN, shoot %N; P.SpM, total spike mass (g). Soil C properties: AFC, aggregate C fraction – %C; C, %C;**
**MFC, MAOM fraction – %C; N, %N; PC, plant-derived C (μg g$^{-1}$ soil); PFC, POM fraction – %C; PRpP, plant-**
**derived C respired proportion; SC, soil-derived C (μg g$^{-1}$ soil); SIC, soil-derived intermediate C (μg C g$^{-1}$ soil); SLC,**
**soil-derived labile C (μg C g$^{-1}$ soil); SRC, soil-derived resistant C (μg C g$^{-1}$ soil); SRpP, soil-derived C respired**
**proportion; TIC, total intermediate C (μg g$^{-1}$ soil); TLC, total labile C (μg g$^{-1}$ soil); TRC, total resistant C (μg g$^{-1}$ soil);**
**TRpP, total C respired proportion.**
**3.4 Fungi-dependent increases in soil C and its stability and persistence are associated with denser fungal hyphae**
Fungal isolates showed strong differentiation in all of the *in vitro*-assessed variables relating to growth and C/N content
(statistically significant effects on all variables, $p < 0.001$; Table B7). Biomass, colony area, and growth rate tended to be
positively associated variables, and were higher in several treatments including *Acrocalymma* sp., *Darksidea* sp. 3, and
*Phialocephala* sp. 1. In contrast, *Thozetella* sp. and *Clohesyomyces* sp. tended to have lower values for these variables, but
*Thozetella* sp. had significantly higher hyphal density than all other treatments.
We performed a separate RDA to identify fungal variables potentially involved in fungi-dependent soil %C and soil C stability
increases, using *in vitro* fungal assessment data and the soil C property data (Fig. 5). Compared to the RDA using plant and
microbial community data (Fig. 4), greater proportions of variance were explained in this RDA by RDA1 and 2 (21.1 and 9%,
respectively). Fungal colony area and hyphal density were close to opposite in their direction, with the high soil C treatment
*Thozetella* sp. closely associated with hyphal density and the low soil C treatment *Clohesyomyces* sp. more associated with
colony area. Similarly, fungal colony maximum growth time and rate (denoting slower and faster fungal growth, respectively)
were in opposing directions. Along this axis, the high soil C treatment *Darksidea* sp. 3 was closely associated with maximum
fungal growth rate. Respired C proportions were closely associated with fungal N content and were opposite resistant C
fractions, which were associated with fungal C/N ratio and hyphal density.



**Figure 5. Fungal isolates involved in increased soil C and its stability have denser hyphae. Redundancy analysis (RDA) showing the fungal variables (blue text) driving changes in soil C properties (red text) associated with the various fungal isolates (symbols). Soil C properties were measured via isotope analysis, soil incubations, and fractionation analysis of soil from wheat experiment. Fungal variables were measured in an *in vitro* plate assay and values were averaged for the RDA. Fungal (F.) variable abbreviations: F.B, biomass (g); F.C, %C; F.CA, final colony area (cm$^2$); F.CN, C/N ratio; F.ECA, estimated final colony area (cm$^2$); F.HD, hyphal density (mg cm$^{-2}$); F.MGR, maximum growth rate (cm$^{-2}$ day); F.MGT, time to maximum growth (days); F.N, %N. Soil C properties: AFC, aggregate C fraction – %C; C,**



**%C; MFC, MAOM fraction – %C; N, %N; PC, plant-derived C (µg g$^{-1}$ soil); PFC, POM fraction – %C; PRpP, plant-derived C respired proportion; SC, soil-derived C (µg g$^{-1}$ soil); SIC, soil-derived intermediate C (µg C g$^{-1}$ soil); SLC, soil-derived labile C (µg C g$^{-1}$ soil); SRC, soil-derived resistant C (µg C g$^{-1}$ soil); SRpP, soil-derived C respired proportion; TIC, total intermediate C (µg g$^{-1}$ soil); TLC, total labile C (µg g$^{-1}$ soil); TRC, total resistant C (µg g$^{-1}$ soil); TRpP, total C respired proportion.**



## 4 Discussion

Discussions on soil C sequestration as a climate change strategy have largely focused on one side of the soil C storage system - increasing C inputs into soil (promoting soil C formation). However, due to the complex and dynamic nature of soil C, reductions of soil C outputs (or, increases in soil C stability and persistence) must also be attained in order to foster soil C storage. In this study, we drew our attention to fungi that have potential in improving soil C storage but that are often overlooked in this area of research, using a high resolution, multifaceted approach combining isotopic labelling, soil incubations, and soil fractionation analysis, as well as an *in vitro* study in parallel. Our study supports the notion that non-mycorrhizal root-associated fungi can improve soil C storage via multiple direct and indirect mechanisms determining C inputs and stabilisation. Mechanisms that increased the stability of existing C were more common across the diverse fungal treatments than those increasing the input of new C.

Despite our finding that bulk soil C increased significantly under only three fungal treatments, in support of our hypothesis our incubations revealed significant increases in directly and functionally assessed soil C stability (i.e. increases in resistant pools and decreases in respired C during incubation) under most of the fungal treatments, with the stabilised C being original soil C, not new inputs of C. Thus, as well as contributing to evidence that fungi can lead to increased soil C content (e.g. Kallenbach et al., 2016), our study provides direct evidence from plant-fungi soil systems for non-mycorrhizal fungi-driven improvements to soil C storage primarily via enhanced stability of soil C. This is emphasised by our findings that the treatments whereby soil C content was the lowest (control, *Clohesyomyces* sp., and *Phialocephala* sp. 1) were associated with higher proportions of total and soil-derived C respired during incubation - indicating that the C remaining at harvest under these treatments was inherently more prone to decomposition (i.e. less persistent). Increased stability and persistence of soil C primarily results from inhibition of microbial decomposition (Cotrufo and Lavallee, 2022), which can occur by a variety of reasons including reduced saprotrophic activity due to microbes being outcompeted for nutrients (Boer et al., 2005), increased input of fungal, more readily stabilised C (Sokol et al., 2019), and increased soil aggregation (Lehmann et al., 2020). We investigated multiple potential mediators for the observed increases in soil C stability and persistence in our study and found some leads. We found that increased fungal C/N ratio and hyphal density may be important for stability and persistence of soil C (while fungal N corresponded with decreased stability and persistence). Fungi with denser hyphae can promote soil aggregation, as soil particles get more entangled and stabilised in dense hyphae (Dignac et al., 2017). Our study substantiates previous assertions that fungal trait expression is relevant to soil C stability: fungi that exhibited an exploitative growth strategy (denser hyphae) were found to more closely associated with soil C stability and persistence, while fungi that exhibited a more exploratory strategy (faster growth) were positively associated with respired C and less stable C pools (Camenzind et al., 2020; Fernandez et al., 2019; Fernandez and Koide, 2013; Jackson et al., 2017; Lehmann et al., 2020; Schmidt et al., 2011; Zanne et al., 2020). These findings support the notion that an exploitative growth strategy may be more conducive to competition with saprotrophs for nutrients, leading to reduced decomposition (Bödeker et al., 2016).





Our PLFA-assessed finding regarding fungal to bacterial ratio points towards a second likely mechanism for the increases in
soil C stability – increased input of fungal C, which becomes necromass. Fungal necromass is a significant source of soil C
inputs, and can in some cases make up the majority of SOM (Wang et al., 2021). Substrates with high C/N ratios, such as
fungal biomass or necromass, are generally associated with reduced decomposition rates, although C/N ratio is not the sole
determinant of substrate decomposition and C/N ratios can in fact be altered by, rather than alter the activity of, soil microbial
communities (Marañón-Jiménez et al., 2021; Smith and Wan, 2019; Schnecker et al., 2019). Compared with other substrates,
however, necromass is a particularly stabilisable form of C as it can bind to the surfaces of MAOM or be stabilised on
aggregates, where it is physically protected from decomposition (Sokol et al., 2019). For these reasons, we expected to see
positive associations between soil C stability and aggregate and MAOM soil fractions, which are considered to signify
increased and longer-term stability and persistence (Dynarski et al., 2020; Hemingway et al., 2019; Islam et al., 2022; Poeplau
et al., 2018; Poeplau et al., 2017). However, in our study these fractions were not strongly associated with soil C content, its
distribution in pools (stability) or persistence, nor were they as influential on differences between fungal treatments. While
this lends support to the notion that microbial decomposition of soil C was metabolically inhibited (as discussed above), rather
than physically limited, our findings may be explained to some extent by methodology. A potential explanation for our findings
is that although fungal necromass may have been abundant, the experimental conditions may have been unsupportive of
MAOM formation (e.g. the high C content of the unplanted soil may have meant that MAOM content was already at saturation
level and new MAOM was not able to form). Other potential explanations are that the MAOM fraction could possibly take
longer than the experimental timeframe to change substantially, or that the MAOM estimation method may carry greater error,
thus making detection of responses more difficult. Nonetheless, our study detected increases in total C, and C resistance and
persistence that were not associated with MAOM, suggesting that soil fractionation analyses do not entirely accurately reflect
natural soil C distribution and stability which can be detected functionally via soil incubations. Further studies utilising the
combined approach of soil incubations and soil fractionation analysis, such as studies using soil with lower C content or studies
over a longer time period, may shed light on how findings from the two methods can be compared. However, our findings call
for caution in directly equating operationally defined MAOM pools and their size with C stability and suggest that functionally
assessing C dynamics may be more effective in some cases.
In terms of improvements to soil C content, of the three fungal treatments whereby soil C increases were significant, these
were accompanied by increases in plant-derived C only under inoculation with *Thozetella* sp. While we expected that there
would be some variation in the fungal impacts on soil C storage due to the diversity amongst the fungi included in this study,
this finding is in contrast to our expectation that increases in plant-derived C would be the main mechanism involved in C
increase. As plant growth promotion and changes in nutrient uptake is a well-known characteristic of some fungi (Hossain et
al., 2017), the increase in plant-derived C with *Thozetella* sp. may have been related to the increases in quantity or quality of
plant inputs related to the shifts in plant variables of *Thozetella* sp. (spike mass, shoot biomass, and shoot C/N ratio). Our
results from the isotopic partitioning of respiration from soil incubations further indicate that the plant-derived C present in




soil and that contributed to total soil C increase under inoculation with *Thozetella* sp. was more persistent compared to the control or other treatments. Fungal-derived C could also have contributed to size and persistence of plant-derived C, if the fungi took up plant-derived C. Thus, in addition to increasing plant inputs, *Thozetella* sp. appears to have been more active in stabilising those inputs via the mechanisms discussed above.

Our study addresses key knowledge gaps in the ways fungi affect soil C storage. We have explicitly demonstrated that inoculation with non-mycorrhizal fungi can improve soil C content and, moreover, soil C stability - supporting the general agreement in this field that microbial transformations of soil C and microbially driven changes to soil structure are as important, if not more important, than the characteristics of the inputs themselves for soil C storage (Dynarski et al., 2020; Hannula and Morriën, 2022). When it comes to evaluating the potential of fungi to support soil C storage, our findings indicate that it is important to consider not only increases in soil C but also their impact on the stability of C. Among the diverse fungi studied, these improvements largely resulted from reductions in C outputs by increasing stable C pools and resistance of existing soil C to decomposition. While potential mechanisms behind these improvements depended on fungal identity, our study points towards metabolic inhibition (rather than physical limitation) of microbial decomposition for which growth characteristics such as density of fungal hyphae and fungal C/N ratio may be important indicators – thus, fungal trait expression may be a proxy for fungal influences on soil C storage. However, more work is needed to test whether or not physical limitation of microbial decomposition leads to enhanced soil C stability by these fungi. More rarely, the improvements to soil C storage involved the effects of fungal inoculation on host plant growth and C inputs. While total soil C content increased significantly only under a minority of fungal treatments, the significant fungi-driven increases in stability we observed could potentially lead to even greater increases in soil C content over time - however experiments with longer timeframes are needed to test this idea. This study and continued work will advance knowledge of these mechanisms and support the search and potential implementation of root-associated fungi to improve soil C storage, which will aid soil C sequestration strategies.



**Appendices**
**Appendix A**



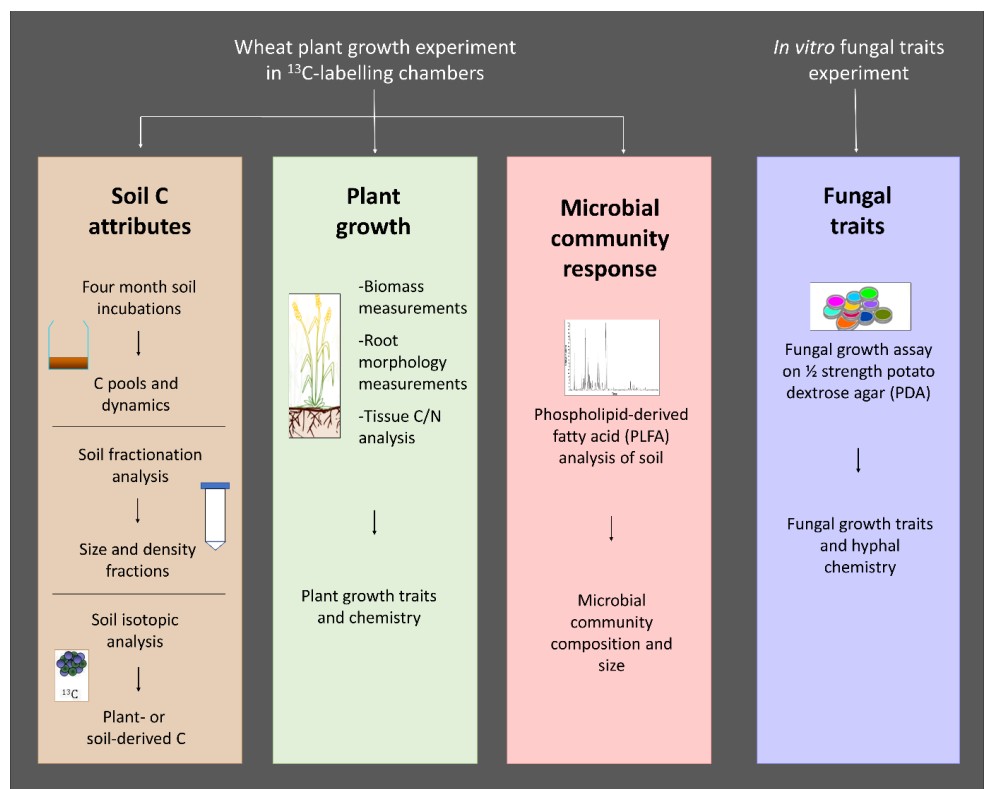

Figure A1. Overview of the study design, measured traits, and methodology used. C, carbon, N, nitrogen.



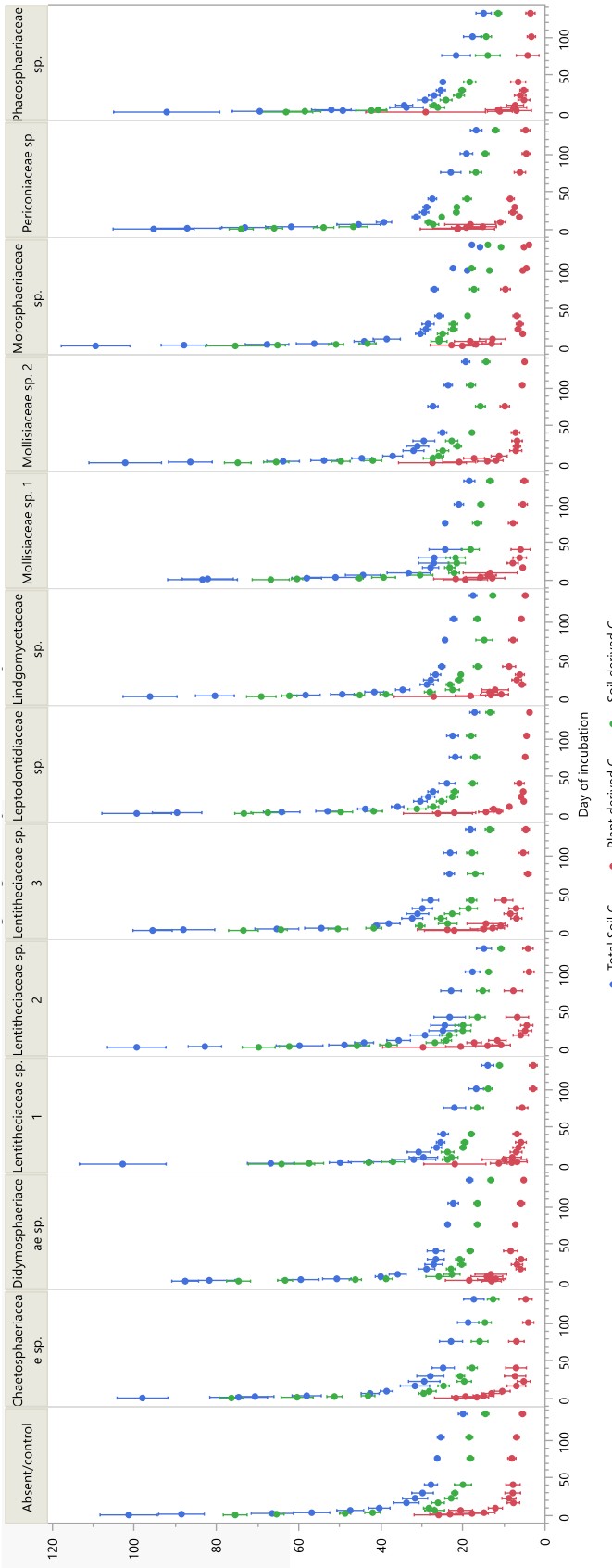

Figure A2. Total soil respiration and its soil- and plant-derived components during laboratory soil incubations of soils collected after plant growth with inoculation of 12 fungal species and a control (Absent/control). Data points are means (n=7 for innoculated pots; n=6 for controls). Soil and plant components calculated from isotopic partitioning based on planted and unplanted soil $\delta^{13}C$. Error bars are standard error.

Family (Genus): Chaetosphaeriaceae sp. (*Thozetella* sp.); Didymosphaeriaceae sp. (*Paraconiothyrium* sp.); Lentitheciaceae sp. 1 (*Darksidea* sp. 1); Lentitheciaceae sp. 2 (*Darksidea* sp. 2); Lentitheciaceae sp. 3 (*Darksidea* sp. 3); Leptodontidiaceae sp. (*Leptodontidium* sp.); Lindgomycetaceae sp. (*Clohesyomyces* sp.); Mollisiaceae sp. 1 (*Phialocephala* sp. 1); Mollisiaceae sp. 2 (*Phialocephala* sp. 2); Morosphaeriaceae sp. (*Acrocalymma* sp.); Periconiaceae sp. (*Periconia* sp.); Phaeosphaeriaceae sp. (*Ophiosphaerella* sp.).





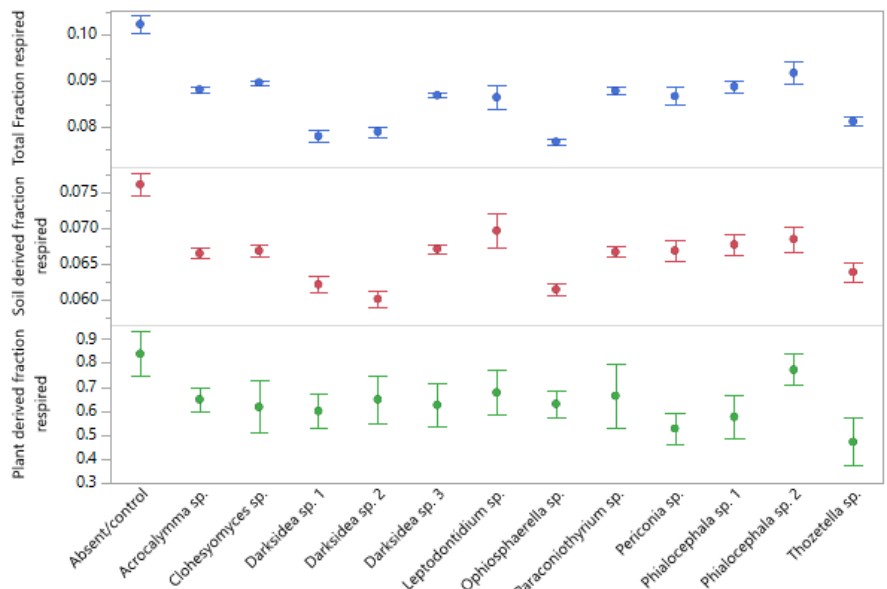

Figure A3. Fraction of soil carbon (C) respired over the course of 135-day incubation of soils under wheat and 12 types of fungal inoculum. Total C is all C respired, and soil- and plant-derived C were obtained from isotopic partitioning of respiration over time (See Materials and methods). Values are means of n=7 for treatments and n=6 for control. Error bars are standard error.



## Appendix B




**Table B1**. Chemical and physical analysis of pre-planted soil used in wheat experiment. Analysis was performed by Environmental Analysis Laboratory (East Lismore, Australia).

| Parameter | Units | Value |
|---|---|---|
| Phosphorus | mg kg$^{-1}$ | 151 |
| pH |  | 5.85 |
| Electrical conductivity | dS m$^{-1}$ | 0.232 |
| Estimated organic matter | % OM | 7.5 |
|  | cmol kg$^{-1}$ | 8.9 |
| Exchangeable calcium | kg ha$^{-1}$ | 4010 |
|  | mg kg$^{-1}$ | 1790 |
|  | cmol kg$^{-1}$ | 2.9 |
| Exchangeable magnesium | kg ha$^{-1}$ | 795 |
|  | mg kg$^{-1}$ | 355 |
|  | cmol kg$^{-1}$ | 3.1 |
| Exchangeable potassium | kg ha$^{-1}$ | 2719 |
|  | mg kg$^{-1}$ | 1214 |
|  | cmol kg$^{-1}$ | 0.32 |
| Exchangeable sodium | kg ha$^{-1}$ | 164 |
|  | mg kg$^{-1}$ | 73 |
|  | cmol kg$^{-1}$ | 0.02 |
| Exchangeable aluminium | kg ha$^{-1}$ | 3.1 |
|  | mg kg$^{-1}$ | 1.4 |
|  | cmol kg$^{-1}$ | 0.06 |
| Exchangeable hydrogen | kg ha$^{-1}$ | 1.2 |
|  | mg kg$^{-1}$ | <1 |
| Effective cation exchange capacity | cmol kg$^{-1}$ | 15 |
| Calcium | % | 58 |
| Magnesium | % | 19 |
| Potassium | % | 20 |
| Exchangeable sodium | % | 2.1 |
| Aluminium | % | 0.1 |
| Hydrogen | % | 0.36 |
| Calcium/magnesium ratio |  | 3.1 |
| Total carbon | % | 4.3 |
| Total nitrogen | % | 0.39 |
| Carbon/nitrogen ratio |  | 11 |
| Basic texture |  | Clay loam |
| Basic colour |  | Brownish |
| Chloride estimate | (equiv. mg kg$^{-1}$) | 148 |



**Table B2.** Properties of soil in which inoculated wheat plants were grown for four months. P-values from ANOVA are displayed in the bottom row. Asterisks/dots in other rows (if present) indicate significant differences to uninoculated controls as determined via Dunnett's post-hoc test ( . p < 0.1, * p < 0.05, ** p < 0.01, *** p < 0.001). C, carbon, N, nitrogen.

| Treatment | %C | %N | $\delta^{13}$C (‰) | $\delta^{15}$N (‰) | Plant-derived C (µg/g soil) | Soil-derived C (µg/g soil) |
|---|---|---|---|---|---|---|
| Absent/control | 3.93 ± 0.07 | 0.36 ± 0.01 | -25.31 ± 0.03 | 9.72 ± 0.04 | 1279.03 ± 247.66 | 38060.63 ± 712.28 |
| *Acrocalymma* sp. | 4.24 ± 0.03 * | 0.39 ± 0.003 ** | -25.33 ± 0.02 | 9.65 ± 0.01 | 1448.55 ± 188.76 | 40966.09 ± 416.19 |
| *Clohesyomyces* sp. | 3.98 ± 0.02 | 0.36 ± 0.003 | -25.33 ± 0.03 | 9.58 ± 0.03 . | 1611.13 ± 319.08 | 38142.72 ± 394.1 |
| *Darksidea* sp. 1 | 4.07 ± 0.06 | 0.37 ± 0.004 | -25.32 ± 0.03 | 9.61 ± 0.06 | 1364.06 ± 220.06 | 39281.97 ± 668.04 |
| *Darksidea* sp. 2 | 4.18 ± 0.06 | 0.38 ± 0.004 . | -25.35 ± 0.03 | 9.62 ± 0.03 | 1635.09 ± 320.66 | 40122.22 ± 683.05 |
| *Darksidea* sp. 3 | 4.23 ± 0.02 * | 0.38 ± 0.003 * | -25.37 ± 0.02 | 9.69 ± 0.02 | 1747.74 ± 243.68 | 40544.37 ± 332.86 |
| *Leptodontidium* sp. | 4.15 ± 0.13 | 0.38 ± 0.01 | -25.34 ± 0.04 | 9.72 ± 0.03 | 1208.67 ± 207.32 | 40246.15 ± 1395.36 |
| *Ophiosphaerella* sp. | 4.11 ± 0.04 | 0.38 ± 0.003 | -25.29 ± 0.04 | 9.82 ± 0.03 | 1004.45 ± 142.31 | 40094.79 ± 501.62 |
| *Paraconiothyrium* sp. | 4.12 ± 0.04 | 0.38 ± 0.004 | -25.39 ± 0.03 | 9.72 ± 0.03 | 1830.47 ± 282.22 | 39356.27 ± 415.96 |
| *Periconia* sp. | 4.18 ± 0.09 | 0.38 ± 0.01 | -25.44 ± 0.04 | 9.75 ± 0.05 | 2038.42 ± 288.09 | 39760.5 ± 820.79 |
| *Phialocephala* sp. 1 | 4.04 ± 0.05 | 0.37 ± 0.01 | -25.36 ± 0.05 | 9.81 ± 0.03 | 1582.66 ± 368.69 | 38769.63 ± 739.07 |
| *Phialocephala* sp. 2 | 4.19 ± 0.10 | 0.38 ± 0.01 * | -25.35 ± 0.02 | 9.71 ± 0.03 | 1422.66 ± 130.89 | 40511.25 ± 998.06 |
| *Thozetella* sp. | 4.30 ± 0.04 ** | 0.39 ± 0.01 ** | -25.47 ± 0.04 * | 9.69 ± 0.03 | 2434.52 ± 418.15 . | 40592.71 ± 756.54 |
| **p-value (ANOVA)** | 0.02 * | 0.01 * | 0.03 * | <0.001 *** | 0.06 . | 0.15 |



**Table B3**. Model fit, model comparisons, pool sizes (resistant, intermediate, and labile) and pool mean residence times (labile and intermediate) estimated from four parameter exponential decay models fitted to $CO_2$ released over 135-day incubations of soil under wheat and fungal inocula. Total C is C in all $CO_2$ released, soil-derived C is C from non-plant origin calculated through isotopic partititioning of $CO_2$ based on plant and $CO_2$ $\delta^{13}$C. Asterisks indicate significant difference with uninoculated controls (. $p < 0.1$, * $p < 0.05$, ** $p < 0.01$, *** $p < 0.001$). Crosses indicate variables for which no statistical test was possible as they were estimated from average curves per treatment. For details of parameter estimation and isotopic partitioning see Materials and methods. C, carbon, MRT, mean residence time.

| | Treatment | Model $R^2$ | Decomposition dynamic p-value (comparison with absent/control group) | Resistant C (µg/g soil) | Intermediate C (µg/g soil)† | Intermediate C MRT (days) | Labile C (µg/g soil)† | Labile C MRT (days) |
|---|---|---|---|---|---|---|---|---|
| **Total C** | Absent/control | 0.89 | NA | 30276 ± 655 | 8777.69 | 247 ± 74 | 285.57 | 3.07 ± 0.40 |
| | *Acrocalymma* sp. | 0.89 | < 0.001 *** | 34923 ± 304 *** | 7195.55 | 210 ± 67 | 295.37 | 2.70 ± 0.33 |
| | *Clohesyomyces* sp. | 0.91 | ns | 31704 ± 206 | 7797.19 | 246 ± 67 | 252.13 | 2.63 ± 0.28 |
| | *Darksidea* sp. 1 | 0.84 | ns | 35164 ± 613 *** | 5275.69 | 164 ± 51 | 206.06 | 1.51 ± 0.22 ** |
| | *Darksidea* sp. 2 | 0.88 | < 0.001 *** | 36182 ± 556 *** | 5322.69 | 160 ± 44 | 252.16 | 2.51 ± 0.37 |
| | *Darksidea* sp. 3 | 0.87 | < 0.01 ** | 34398 ± 195 ** | 7620.96 | 222 ± 65 | 272.88 | 3.01 ± 0.42 |
| | *Leptodontidium* sp. | 0.89 | ns | 33941 ± 1285 ** | 7216.05 | 227 ± 69 | 297.45 | 3.04 ± 0.37 |
| | *Ophiosphaerella* sp. | 0.79 | ns | 35583 ± 380 *** | 5317.96 | 161 ± 60 | 198.12 | 2.09 ± 0.45 |
| | *Paraconiothyrium* sp. | 0.89 | ns | 32053 ± 379 | 8866.63 | 291 ± 97 | 266.34 | 3.25 ± 0.41 |
| | *Periconia* sp. | 0.87 | ns | 34970 ± 859 *** | 6485.94 | 196 ± 77 | 342.66 | 4.17 ± 0.81 |
| | *Phialocephala* sp. 1 | 0.79 | < 0.001 *** | 31058 ± 540 | 9011.62 | 309 ± 193 | 282.05 | 3.76 ± 0.77 |
| | *Phialocephala* sp. 2 | 0.88 | < 0.01 ** | 33098 ± 1041. | 8563.14 | 249 ± 79 | 271.87 | 2.73 ± 0.35 |
| | *Thozetella* sp. | 0.86 | < 0.001 *** | 36615 ± 439 *** | 6127.71 | 182 ± 54 | 284.05 | 3.41 ± 0.53 |
| **Soil-derived C** | Absent/control | 0.95 | NA | 31337 ± 712 | 6517.67 | 258± 55 | 205.43 | 2.70 ± 0.22 |
| | *Acrocalymma* sp. | 0.9 | < 0.001 *** | 35086 ± 416 * | 5660.13 | 234 ± 77 | 219.30 | 2.90 ± 0.34 |
| | *Clohesyomyces* sp. | 0.94 | ns | 32351 ± 394 | 5586.36 | 252 ± 60 | 205.31 | 2.99 ± 0.25 |
| | *Darksidea* sp. 1 | 0.85 | ns | 34436 ± 668. | 4669.97 | 206 ± 75 | 175.08 | 2.78 ± 0.43 |
| | *Darksidea* sp. 2 | 0.92 | < 0.001 *** | 35757 ± 683 ** | 4165.06 | 181 ± 45 | 199.37 | 2.86 ± 0.33 |
| | *Darksidea* sp. 3 | 0.93 | < 0.001 *** | 33927 ± 332 | 6389.46 | 277 ± 78 | 227.75 | 3.18 ± 0.30 |
| | *Leptodontidium* sp. | 0.92 | < 0.001 *** | 34232 ± 1395 | 5791.95 | 235 ± 58 | 221.83 | 3.13 ± 0.32 |
| | *Ophiosphaerella* sp. | 0.87 | ns | 35804 ± 501 ** | 4113.89 | 169 ± 52 | 175.91 | 3.10 ± 0.56 |
| | *Paraconiothyrium* sp. | 0.95 | ns | 32887 ± 415 | 6258.33 | 281 ± 64 | 209.99 | 2.64 ± 0.19 |
| | *Periconia* sp. | 0.96 | ns | 34874 ± 820 * | 4644.09 | 187 ± 37 | 242.11 | 3.58 ± 0.34 * |
| | *Phialocephala* sp. 1 | 0.91 | < 0.001 *** | 32988 ± 739 | 5584.94 | 241 ± 74 | 196.62 | 3.14 ± 0.38 |
| | *Phialocephala* sp. 2 | 0.93 | < 0.001 *** | 33891 ± 998 | 6399.73 | 270 ± 72 | 220.25 | 2.94 ± 0.27 |
| | *Thozetella* sp. | 0.94 | < 0.001 *** | 35864 ± 756 ** | 4509.96 | 184 ± 37 | 217.77 | 3.05 ± 0.29 |



**Table B4.** Properties of carbon fractions of soil in which inoculated wheat plants were grown for four months. Properties were measured using soil fractionation analysis. P-values from ANOVA are displayed in the bottom row. Asterisks/dots in other rows (if present) indicate significant differences to uninoculated controls as determined via Dunnett's post-hoc test (., p < 0.1, * p < 0.05, ** p < 0.01, *** p < 0.001). C, carbon, N, nitrogen, AggC, aggregate carbon, MAOM, mineral-associated organic matter, POM, particulate organic matter.

| Treatment | AggC fraction – %C | AggC fraction – %N | MAOM fraction – %C | MAOM fraction – %N | POM fraction – %C | POM fraction – %N |
|---|---|---|---|---|---|---|
| Absent/control | 1.96 ± 0.05 | 0.16 ± 0.01 | 0.57 ± 0.02 | 0.05 ± 0.002 | 0.92 ± 0.07 | 0.06 ± 0.01 |
| *Acrocalymma* sp. | 2.18 ± 0.10 | 0.18 ± 0.01 | 0.48 ± 0.02 | 0.04 ± 0.001 | 0.98 ± 0.05 | 0.07 ± 0.004 |
| *Clohesyomyces* sp. | 2.14 ± 0.07 | 0.18 ± 0.01 | 0.51 ± 0.02 | 0.05 ± 0.002 | 0.94 ± 0.05 | 0.06 ± 0.003 |
| *Darksidea* sp. 1 | 2.09 ± 0.06 | 0.17 ± 0.01 | 0.58 ± 0.04 | 0.05 ± 0.003 | 0.87 ± 0.04 | 0.06 ± 0.003 |
| *Darksidea* sp. 2 | 2.13 ± 0.03 | 0.17 ± 0.002 | 0.54 ± 0.05 | 0.05 ± 0.004 | 0.89 ± 0.03 | 0.06 ± 0.002 |
| *Darksidea* sp. 3 | 2.13 ± 0.05 | 0.17 ± 0.004 | 0.60 ± 0.02 | 0.05 ± 0.002 | 1.00 ± 0.06 | 0.07 ± 0.004 |
| *Leptodontidium* sp. | 2.12 ± 0.07 | 0.17 ± 0.01 | 0.53 ± 0.02 | 0.05 ± 0.002 | 0.98 ± 0.04 | 0.06 ± 0.003 |
| *Ophiosphaerella* sp. | 2.18 ± 0.04 | 0.19 ± 0.004 * | 0.55 ± 0.03 | 0.05 ± 0.003 | 0.96 ± 0.04 | 0.07 ± 0.003 |
| *Paraconiothyrium* sp. | 2.15 ± 0.05 | 0.18 ± 0.004 | 0.56 ± 0.03 | 0.05 ± 0.002 | 1.00 ± 0.06 | 0.07 ± 0.01 |
| *Periconia* sp. | 2.25 ± 0.06 * | 0.19 ± 0.01 * | 0.55 ± 0.05 | 0.05 ± 0.004 | 0.89 ± 0.03 | 0.06 ± 0.002 |
| *Phialocephala* sp. 1 | 2.22 ± 0.06 | 0.19 ± 0.01 ** | 0.53 ± 0.02 | 0.05 ± 0.002 | 0.86 ± 0.09 | 0.06 ± 0.01 |
| *Phialocephala* sp. 2 | 2.09 ± 0.07 | 0.17 ± 0.01 | 0.56 ± 0.03 | 0.05 ± 0.003 | 0.86 ± 0.03 | 0.06 ± 0.002 |
| *Thozetella* sp. | 2.37 ± 0.07 *** | 0.20 ± 0.01 *** | 0.52 ± 0.04 | 0.05 ± 0.003 | 0.91 ± 0.10 | 0.06 ± 0.01 |
| **p-value (ANOVA)** | 0.03 * | 0.002 ** | 0.63 | 0.62 | 0.65 | 0.41 |





**Table B5.** Plant variables potentially influencing soil (in which inoculated wheat plants were grown for four months). P-values from ANOVA are displayed in bottom rows. Asterisks/dots in other rows (if present) indicate significant differences to uninoculated controls as determined via Dunnett's post-hoc test (. p < 0.1, * p < 0.05, ** p < 0.01, *** p < 0.001). C, carbon, N, nitrogen.

| Treatment | Number of spikes | Average spike mass (g) | Total spike mass (g) | Shoot mass (g) | Root mass (g) | Root/shoot ratio | Shoot $\delta^{13}C$ (‰) | Shoot $\delta^{15}N$ (‰) | Shoot %C |
|---|---|---|---|---|---|---|---|---|---|
| Absent/control | 5.50 ± 0.91 | 1.52 ± 0.28 | 7.36 ± 1.06 | 16.38 ± 1.97 | 2.23 ± 0.20 | 0.14 ± 0.01 | -32.27 ± 0.92 | 9.74 ± 0.24 | 38.30 ± 0.42 |
| *Acrocalymma* sp. | 4.86 ± 0.43 | 1.82 ± 0.07 | 8.81 ± 0.81 | 16.81 ± 1.77 | 1.83 ± 0.33 | 0.11 ± 0.01 | -32.47 ± 0.91 | 9.39 ± 0.15 | 37.81 ± 0.40 |
| *Clohesyomyces* sp. | 4.14 ± 0.65 | 1.85 ± 0.25 | 6.60 ± 0.77 | 13.28 ± 1.26 | 1.44 ± 0.22 | 0.11 ± 0.01 | -31.94 ± 1.02 | 9.38 ± 0.18 | 38.21 ± 0.49 |
| *Darksidea* sp. 1 | 3.86 ± 0.24 | 2.13 ± 0.10 | 8.11 ± 0.38 | 15.54 ± 0.95 | 1.75 ± 0.17 | 0.11 ± 0.01 | -32.27 ± 1.03 | 9.44 ± 0.18 | 38.07 ± 0.28 |
| *Darksidea* sp. 2 | 4.43 ± 0.45 | 2.20 ± 0.14 | 9.41 ± 0.68 | 16.88 ± 1.55 | 2.00 ± 0.25 | 0.12 ± 0.01 | -32.19 ± 0.84 | 9.64 ± 0.34 | 38.08 ± 0.49 |
| *Darksidea* sp. 3 | 4.14 ± 0.84 | 1.63 ± 0.20 | 6.37 ± 1.17 | 15.46 ± 1.62 | 1.86 ± 0.34 | 0.14 ± 0.02 | -32.73 ± 1.13 | 9.89 ± 0.13 | 37.72 ± 0.52 |
| *Leptodontidium* sp. | 5.57 ± 0.90 | 1.72 ± 0.25 | 8.15 ± 0.66 | 16.42 ± 0.80 | 2.02 ± 0.44 | 0.12 ± 0.03 | -33.53 ± 0.76 | 9.21 ± 0.48 | 37.73 ± 0.59 |
| *Ophiosphaerella* sp. | 4.43 ± 0.28 | 1.92 ± 0.11 | 8.32 ± 0.26 | 15.68 ± 1.17 | 1.63 ± 0.40 | 0.10 ± 0.02 | -32.76 ± 1.08 | 9.37 ± 0.24 | 37.57 ± 0.32 |
| *Paraconiothyrium* sp. | 3.86 ± 0.51 | 2.12 ± 0.23 | 7.43 ± 0.40 | 14.01 ± 1.03 | 1.73 ± 0.35 | 0.12 ± 0.02 | -32.32 ± 0.95 | 9.66 ± 0.38 | 37.21 ± 0.36 |
| *Periconia* sp. | 3.86 ± 0.51 | 1.93 ± 0.20 | 7.36 ± 1.07 | 15.96 ± 1.48 | 1.83 ± 0.23 | 0.12 ± 0.02 | -32.42 ± 0.86 | 10.23 ± 0.26 | 38.17 ± 0.32 |
| *Phialocephala* sp. 1 | 4.43 ± 0.60 | 1.98 ± 0.25 | 7.85 ± 0.60 | 15.82 ± 1.34 | 1.93 ± 0.36 | 0.12 ± 0.02 | -32.42 ± 0.96 | 9.15 ± 0.16 | 38.43 ± 0.35 |
| *Phialocephala* sp. 2 | 4.00 ± 0.54 | 2.26 ± 0.20 | 8.56 ± 0.85 | 15.95 ± 1.90 | 2.19 ± 0.28 | 0.14 ± 0.01 | -32.68 ± 0.86 | 9.80 ± 0.19 | 37.64 ± 0.33 |
| *Thozetella* sp. | 4.14 ± 0.51 | 2.48 ± 0.15 * | 9.82 ± 0.66 | 18.57 ± 1.55 | 2.55 ± 0.36 | 0.14 ± 0.02 | -32.58 ± 1.07 | 9.31 ± 0.23 | 37.66 ± 0.41 |
| **p-value (ANOVA)** | 0.66 | 0.12 | 0.14 | 0.14 | 0.75 | 0.74 | 0.82 | 1.00 | 0.84 |

| Treatment | Shoot %N P.SN | Shoot C/N ratio P.SCN | Root length density (cm/g) P.RLDe | Root specific surface area (cm²/g) P.RSA | Root average diameter (mm) P.RADi | Root length per volume (cm/m³) P.RLV | Root specific density (g/cm³) P.RSDe | Root fork number (/g) P.RF |
|---|---|---|---|---|---|---|---|---|
| Absent/control | 0.49 ± 0.05 | 83.32 ± 8.44 | 3315.39 ± 307.45 | 490.13 ± 30.83 | 0.48 ± 0.02 | 515.85 ± 65.777 | 0.17 ± 0.01 | 5878.38 ± 870.62 |
| *Acrocalymma* sp. | 0.43 ± 0.03 | 90.51 ± 7.10 | 3563.82 ± 247.20 | 530.07 ± 31.47 | 0.48 ± 0.01 | 492.79 ± 95.89 | 0.16 ± 0.01 | 6456.09 ± 1283.54 |
| *Clohesyomyces* sp. | 0.45 ± 0.04 | 91.07 ± 7.69 | 4044.30 ± 627.70 | 561.07 ± 63.37 | 0.46 ± 0.03 | 499.66 ± 102.50 | 0.17 ± 0.01 | 7056.00 ± 1385.96 |
| *Darksidea* sp. 1 | 0.44 ± 0.04 | 90.30 ± 6.73 | 3544.01 ± 390.12 | 539.47 ± 52.13 | 0.49 ± 0.02 | 586.57 ± 61.95 | 0.16 ± 0.01 | 6748.77 ± 1228.20 |
| *Darksidea* sp. 2 | 0.40 ± 0.02 | 97.22 ± 6.10 | 3872.21 ± 461.38 | 557.82 ± 39.54 | 0.48 ± 0.02 | 620.39 ± 123.60 | 0.16 ± 0.01 | 8050.86 ± 1549.33 |
| *Darksidea* sp. 3 | 0.58 ± 0.12 | 82.65 ± 12.54 | 3912.67 ± 356.62 | 562.39 ± 27.00 | 0.47 ± 0.02 | 570.09 ± 136.56 | 0.15 ± 0.01 | 7540.25 ± 1301.61 |
| *Leptodontidium* sp. | 0.46 ± 0.04 | 85.82 ± 6.59 | 3779.06 ± 475.55 | 540.19 ± 41.41 | 0.47 ± 0.03 | 615.66 ± 145.93 | 0.16 ± 0.01 | 6972.52 ± 1670.66 |
| *Ophiosphaerella* sp. | 0.43 ± 0.02 | 89.68 ± 5.32 | 4718.73 ± 906.96 | 632.58 ± 83.92 | 0.45 ± 0.02 | 698.43 ± 146.81 | 0.15 ± 0.01 | 9458.82 ± 2376.20 |
| *Paraconiothyrium* sp. | 0.44 ± 0.05 | 93.43 ± 10.56 | 3721.05 ± 352.69 | 541.97 ± 40.66 | 0.47 ± 0.02 | 440.31 ± 85.04 | 0.16 ± 0.01 | 6278.34 ± 1226.28 |
| *Periconia* sp. | 0.59 ± 0.11 | 75.07 ± 8.24 | 3629.11 ± 390.34 | 520.13 ± 38.44 | 0.47 ± 0.02 | 465.06 ± 89.46 | 0.17 ± 0.01 | 6273.79 ± 1414.99 |
| *Phialocephala* sp. 1 | 0.41 ± 0.03 | 96.97 ± 7.95 | 3170.61 ± 220.70 | 469.51 ± 30.03 | 0.47 ± 0.01 | 382.08 ± 67.80 | 0.19 ± 0.01 | 4430.48 ± 488.78 |
| *Phialocephala* sp. 2 | 0.45 ± 0.05 | 91.12 ± 9.15 | 4648.09 ± 804.77 | 631.31 ± 76.97 | 0.45 ± 0.02 | 748.74 ± 106.18 | 0.15 ± 0.01 | 9350.21 ± 1855.27 |
| *Thozetella* sp. | 0.39 ± 0.03 | 99.44 ± 7.41 | 3651.81 ± 353.05 | 521.36 ± 30.21 | 0.47 ± 0.02 | 697.98 ± 92.43 | 0.17 ± 0.01 | 6835.67 ± 1146.69 |
| **p-value (ANOVA)** | 0.47 | 0.86 | 0.75 | 0.75 | 0.68 | 0.10 | 0.98 | 0.55 | 0.69 |



**Table B6.** Microbial community variables potentially influencing soil (in which inoculated wheat plants were grown for four months). P-values from ANOVA are displayed in the bottom row. Asterisks/dots in other rows (if present) indicate significant differences to uninoculated controls as determined via Dunnett's post-hoc test ( p < 0.1, * p < 0.05, ** p < 0.01, *** p < 0.001).

| Treatment | Total community size (µg PLFA/g soil) | Fungal to bacterial biomass ratio | Gram positive bacteria (% of total community) | Gram negative bacteria (% of total community) | Actinobacteria (% of total community) | Fungi (% of total community) | Arbuscular mycorrhizal fungi (% of total community) |
|---|---|---|---|---|---|---|---|
| Absent/control | 8.30 ± 0.33 | 0.22 ± 0.01 | 19.50 ± 0.01 | 26.19 ± 0.55 | 8.20 ± 0.14 | 10.19 ± 0.47 | 2.41 ± 0.09 |
| *Acrocalymma* sp. | 8.59 ± 0.57 | 0.23 ± 0.01 | 19.88 ± 0.01 | 26.10 ± 0.72 | 7.68 ± 0.74 | 10.44 ± 0.42 | 2.45 ± 0.07 |
| *Clohesyomyces* sp. | 8.35 ± 0.28 | 0.22 ± 0.01 | 20.38 ± 0.01 | 26.48 ± 0.48 | 8.48 ± 0.14 | 10.11 ± 0.28 | 2.52 ± 0.07 |
| *Darksidea* sp. 1 | 8.54 ± 0.30 | 0.22 ± 0.01 | 20.14 ± 0.01 | 26.06 ± 0.61 | 8.37 ± 0.11 | 9.98 ± 0.26 | 2.63 ± 0.10 |
| *Darksidea* sp. 2 | 7.72 ± 0.32 | 0.21 ± 0.01 | 20.10 ± 0.01 | 26.59 ± 0.47 | 8.23 ± 0.16 | 9.79 ± 0.32 | 2.71 ± 0.12 |
| *Darksidea* sp. 3 | 7.50 ± 0.71 | 0.22 ± 0.01 | 19.03 ± 0.01 | 25.32 ± 0.40 | 7.90 ± 0.08 | 9.54 ± 0.34 | 2.41 ± 0.08 |
| *Leptodontidium* sp. | 7.89 ± 0.51 | 0.23 ± 0.01 | 20.01 ± 0.01 | 26.02 ± 0.57 | 8.16 ± 0.20 | 10.36 ± 0.41 | 2.62 ± 0.07 |
| *Ophiosphaerella* sp. | 8.61 ± 0.21 | 0.24 ± 0.01 | 19.28 ± 0.01 | 26.27 ± 0.33 | 8.21 ± 0.17 | 10.97 ± 0.47 | 2.72 ± 0.08 |
| *Paraconiothyrium* sp. | 7.98 ± 0.27 | 0.21 ± 0.01 | 20.65 ± 0.01 | 26.64 ± 0.43 | 8.69 ± 0.15 | 9.88 ± 0.29 | 2.65 ± 0.05 |
| *Periconia* sp. | 8.50 ± 0.34 | 0.21 ± 0.01 | 20.37 ± 0.01 | 27.02 ± 0.34 | 8.25 ± 0.09 | 9.83 ± 0.34 | 2.61 ± 0.09 |
| *Phialocephala* sp. 1 | 8.69 ± 0.29 | 0.21 ± 0.01 | 20.52 ± 0.01 | 26.34 ± 0.42 | 8.30 ± 0.09 | 9.79 ± 0.27 | 2.75 ± 0.09 . |
| *Phialocephala* sp. 2 | 8.75 ± 0.20 | 0.23 ± 0.01 | 19.30 ± 0.01 | 25.89 ± 0.27 | 8.25 ± 0.19 | 10.16 ± 0.43 | 2.62 ± 0.09 |
| *Thozetella* sp. | 8.27 ± 0.37 | 0.22 ± 0.01 | 19.39 ± 0.01 | 26.23 ± 0.50 | 8.23 ± 0.11 | 9.80 ± 0.24 | 2.53 ± 0.09 |
| **p-value (ANOVA)** | 0.72 | 0.50 | 0.45 | 0.81 | 0.61 | 0.50 | 0.13 |





**Table B7.** Fungal variables potentially influencing soil (in which inoculated wheat plants were grown for four months). P-values from ANOVA are displayed in the bottom row ( $p < 0.1$, * $p < 0.05$, ** $p < 0.01$, *** $p < 0.001$). Different letters indicate significant differences between treatments as determined via Tukey's post-hoc test. † indicates variables calculated using treatment averages. C, carbon, N, nitrogen.

| Treatment | Estimated final colony area (cm²)† | Maximum growth rate (cm²/day)† | Time to maximum growth (days)† | Biomass (g)† | Final colony area (cm²)† | Hyphal density (mg/cm²)† | %C† | %N† | C/N ratio† |
|---|---|---|---|---|---|---|---|---|---|
| *Acrocalymma* sp. | 53.58 ± 1.26 c | 4.61 ± 0.03 de | 12.02 ± 0.26 bcd | 0.12 ± 0.01 ab | 49.17 ± 0.55 abc | 2.42 ± 0.23 b | 51.96 ± 0.37 ab | 2.67 ± 0.06 cd | 19.53 ± 0.36 bc |
| *Clohesyomyces* sp. | 38.64 ± 1.72 d | 2.05 ± 0.08 g | 17.42 ± 0.28 a | 0.04 ± 0.01 e | 29.76 ± 1.78 d | 1.18 ± 0.23 b | 49.11 ± 0.49 cd | 3.81 ± 0.09 a | 12.93 ± 0.41 f |
| *Darksidea* sp. 1 | 59.49 ± 1.94 bc | 3.39 ± 0.09 f | 18.04 ± 0.36 a | 0.08 ± 0.003 cd | 47.43 ± 1.14 bc | 1.61 ± 0.09 b | 45.99 ± 0.23 e | 2.32 ± 0.07 de | 19.91 ± 0.57 bc |
| *Darksidea* sp. 2 | 69.82 ± 0.84 ab | 4.89 ± 0.09 cd | 16.87 ± 0.09 a | 0.09 ± 0.01 bcd | 53.58 ± 0.96 ab | 1.70 ± 0.12 b | 46.96 ± 0.18 e | 2.55 ± 0.10 d | 18.53 ± 0.77 cd |
| *Darksidea* sp. 3 | 58.39 ± 1.04 bc | 5.12 ± 0.06 cd | 12.93 ± 0.10 bc | 0.07 ± 0.004 cde | 52.52 ± 0.63 ab | 1.35 ± 0.08 b | 52.81 ± 0.30 a | 2.66 ± 0.04 cd | 19.91 ± 0.35 bc |
| *Leptodontidium* sp. | 53.01 ± 2.42 c | 4.00 ± 0.21 ef | 16.20 ± 0.20 a | 0.08 ± 0.01 cde | 43.02 ± 2.40 c | 1.80 ± 0.23 b | 52.68 ± 0.32 a | 2.06 ± 0.03 e | 25.54 ± 0.28 a |
| *Ophiosphaerella* sp. | 70.45 ± 1.50 ab | 6.37 ± 0.02 b | 13.63 ± 0.22 b | 0.13 ± 0.01 a | 54.45 ± 0.24 a | 2.44 ± 0.24 b | 50.42 ± 0.52 bc | 2.09 ± 0.03 e | 24.16 ± 0.03 a |
| *Paraconiothyrium* sp. | 74.83 ± 3.68 a | 7.54 ± 0.11 a | 10.19 ± 0.27 de | 0.09 ± 0.01 abcd | 50.25 ± 0.67 ab | 1.86 ± 0.15 b | 47.43 ± 0.46 de | 3.02 ± 0.15 bc | 15.83 ± 0.66 e |
| *Periconia* sp. | 66.92 ± 2.66 ab | 7.28 ± 0.04 a | 9.81 ± 0.32 e | 0.09 ± 0.004 bcd | 48.01 ± 0.41 abc | 1.82 ± 0.09 b | 52.54 ± 0.17 a | 3.24 ± 0.07 b | 16.24 ± 0.17 de |
| *Phialocephala* sp. 1 | 60.76 ± 2.03 bc | 5.35 ± 0.17 c | 13.51 ± 0.15 bc | 0.10 ± 0.003 abcd | 53.34 ± 1.43 ab | 1.87 ± 0.08 b | 46.51 ± 0.19 e | 2.38 ± 0.02 de | 19.58 ± 0.26 bc |
| *Phialocephala* sp. 2 | 58.61 ± 1.74 abc | 5.12 ± 0.06 cd | 12.32 ± 0.16 bcde | 0.12 ± 0.01 abc | 53.46 ± 1.10 ab | 2.15 ± 0.13 b | 45.87 ± 0.44 e | 2.30 ± 0.02 de | 19.98 ± 0.14 bc |
| *Thozetella* sp. | 28.02 ± 4.16 d | 2.16 ± 0.19 g | 11.33 ± 1.05 cde | 0.06 ± 0.01 de | 13.95 ± 1.17 e | 4.59 ± 0.54 a | 50.97 ± 0.35 abc | 2.42 ± 0.02 de | 21.10 ± 0.35 b |
| p-value (ANOVA) | <0.001 *** | <0.001 *** | <0.001 *** | <0.001 *** | <0.001 *** | <0.001 *** | <0.001 *** | <0.001 *** | <0.001 *** |



**Author contribution**

YC, JP and LCG designed the study; ES, LCG and WB performed the research; ES wrote the first draft of the manuscript, and all authors contributed to revisions.

**Competing interests**

The research was partially funded by SoilCQuest2031 who provided the fungal cultures and soil. This funding was provided independently of research findings. SoilCQuest2031 did not attempt to influence the interpretations or conclusions of the work. The authors declare that the research was conducted in the absence of any commercial or financial relationships that could be construed as a potential conflict of interest.

**Acknowledgements**

This project was supported by Western Sydney University Research Partnership Program and by SoilCQuest2031 (Orange, New South Wales, Australia). We acknowledge assistance from Guy Webb and Suresh Subashchandrabose for providing soils and cultures. We also thank Andrew Gherlenda for assistance with the growth chamber experiment, Russell Thomson for help with nonlinear least square curve fitting, UC Davis Stable Isotope Facility, Environmental Analysis Laboratory, and Pushpinder Matta for running the nutrient analyses, and Sophia Bruna, Hui Zhang, and Asel Weerasekara for help with the experiment harvest.



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
