# Peer review of "Non-mycorrhizal root-associated fungi increase soil C stocks and stability via diverse mechanisms"

_EGUsphere, 2023_

## Author Response (AR2)

We thank the reviewers and associate editor for their constructive comments on our manuscript and the opportunity to respond to their comments. Please find below a point by point response to each comment and the line numbers of the tracked changes document where the edits were made.

Reviewer 1:

General comments:

This is a very well-written and clear description of a great study. I enjoyed reading it and I think the results will be of interest to many.

One big-picture comment I have is about the fungal isolates used here—are they representative of fungal symbionts on wheat plants in general? Does that impact the way we can interpret these results? Or was this study more of a proof of concept, and future work should be done with more ecologically relevant plant/fungal partnerships?

Response:

The fungal isolates used in our study are not representative of wheat root symbionts, as they were isolated from natural and diverse environments as part of a bioprospecting effort. This was done with the specific aim to identify novel organisms that can be introduced broadly to crop plants to improve soil C accrual. We have made changes to better emphasise this point in the Abstract (L10), Introduction (L100-101) and add more details of the selection process to the Materials and methods (L122-129).

Secondly, in nature we know there will be multiple fungal symbionts in the plant roots, including non-mycorrhizal and mycorrhizal fungi. Are there any existing data to suggest that these combinations might influence the overall effect on soil C—as in, one species of root-associated fungi counteracts the effects of another? Or might they add to one another?

These are ideas that might be worth mentioning in the discussion or possibly the introduction.

Response:

This is a good point - we agree that interactions with other root symbionts likely impact the effects of the fungal isolates used in this study and broadly in nature. In this study, we assessed the net outcomes of fungal inoculation; thus, if interactions with other endophytes occurred and we still observed impacts, the interpretation would be that particular isolates are able to have a net effect. Likewise, the absence of an effect may have resulted from an interaction with another organism. We have made changes to the Discussion, and added emphasis throughout the paper that we are looking at net outcomes of fungal inoculation (L95 and L462-464).

Specific comments:

Line 81-83 "However…" : I would suggest these conflicting reports are just as common for AM and ECM fungi! All the more reason non-mycorrhizal plant associates should be considered as a potentially equally important group.

Response:

Thank you for raising this point. We have edited the wording in this paragraph to highlight the fact that there are conflicting reports for even the well-studied AM and ECM fungi, which further necessitates the research in this area (L81-92).

Line 115: isolated from what kind of plant roots; wheat, or another species? If another species, how was it determined that these would inoculate the wheat successfully?

Response:

The fungal isolates were isolated from surface-sterilised roots of multiple plant species, primarily shrubs and grasses, across diverse natural environments in southeast Australia. These species included: *Chloris truncata*, *Paspalum* sp., *Poa sieberiana*, *Austrostipa* sp., and *Enchylaena tomentosa*. The screening process involved inoculating crop plants (including wheat) with individual isolates to determine successful colonisation. We have added further details to the Materials and methods (L122-129).

Line 116: What traits were looked at in this screening process? Can you give more information about the nature of the screening?

Response:

In addition to testing for successful colonisation, tests were done for soil C increases, pathogenicity, fungicide compatibility, P solubilisation, interaction with other bacteria and fungi, and resilience in environmental fluctuations (pH, moisture, salinity, temperature). Economic viability was also considered. We have added further information on the screening process to the Materials and methods (L122-129).

Line 121: Was the soil used in the pot experiment previously used to grow wheat, or another crop? Was it sterilized?

Response:

The past 10 years of land use history for the soil included wheat, barley, canola, and sorghum. The soil was not sterilised. We have added this to the Materials and methods (L132-133).

Line 125: What concentration/rate of 13CO2 was added to the chambers?

Response:

$^{13}$C depleted $CO_2$ was added to the chamber at a rate to maintain the target $CO_2$ concentration of 450 ppm (as dictated by the chambers' $CO_2$ concentration controls). We have added this to the Materials and methods (L141-143).

Line 134 (and 96) I take it you mean 13C enriched?

Response:

We utilised the $^{13}$C depletion method (rather than the enrichment method) to label plant tissues with $^{13}$C. We have added the following details to the Materials and methods for further clarity (L139-143).

"The $CO_2$-controlled growth chambers were modified using the approach by Cheng and Dijkstra (2007) to achieve continuous $^{13}$C-labeling of plant tissues. Briefly, the chambers were adapted to take an influx of naturally $^{13}$C-depleted $CO_2$ ($\delta^{13}$C = -31.7 o/oo ± 1.2) during the photoperiod, combined with a continuous supply of external $CO_2$-free air, and set to 450 ppm $CO_2$ concentration."

Line 204: How many missing data points were there?

Response:

One for root mass, two for shoot mass and root/shoot ratio, 12 for fraction of respired plant C (removed values >1 as fraction could not be calculated). We have added details to the text (L230).

Line 201: How many ANOVAs were performed? Was any p value adjustment made for multiple tests?

Response:

ANOVAs were performed for the soil properties (Table B2 and B4), plant variables (Table B5), microbial community (Table B6), and fungal properties (Table B7). For multiple comparisons we used Dunnett's test to compare with the control and Tukey's test for comparing between isolates. Both of these control for inflation of Type I error.

Line 203-205: What were the PCA and RDA tests for? (What variables were included, what questions were those tests addressing, etc.)

We have included this information by changing this sentence to the following (L225-230):

"Principal component analysis (PCA) of soil C property data was performed to identify soil C properties associated with fungi-driven increases in soil C. Redundancy analyses (RDA) of soil C property data as response variables and either plant and microbial community data or using in vitro fungal assessment data as explanatory variables were performed to identify explanatory variables for fungi-driven increases in soil C and its stability. Both analyses were performed using the vegan package in R (Oksanen et al., 2020)."

Line 208: Are these scripts available to the public?

Response:

We will make these available according to the journal's policies when the manuscript is accepted for publication.

Figure 1: This appears to be a figure from JMP, but this analysis was performed in R, correct? I would recommend using R to generate this figure with the ggplot2 syntax, which will allow you to produce a higher resolution image with more control over the appearance. The basic syntax for a plot like this in ggplot would be something like:

plot=ggplot(data, aes(x=Isolate, y=ChangeInSoilC))+

  geom_point()+

  stat_summary(fun= mean, fun.min=mean, fun.max=mean, geom="crossbar", width=0.5)+

stat_summary(fun.data = mean_se, geom = "errorbar", width=0.2)+

  theme_bw()+ #simplifies the aesthetics

  facet_grid(~PlantedOrUnplanted, nrow=1)

Response:

The means comparisons were performed in JMP. The lack of resolution is due the formatting for pasting. We have improved the resolution of this figure.

Line 231: Please indicate the test statistic and p-value for this result.

Response:

The t-value is 4.13 and p-value is < 0.001. We have added this information to the text (L258).

Line 232-233: I would either edit this statement or add another one making it clear that there were three fungal treatments that showed an increase in total soil C compared with uninoculated pots.

Response:

The statement in line 235-236 makes it clear that there were three fungal treatments with soil C increases. To make this statement less confusing, we have changed this to "Soil C increases under fungal inoculation had different origins depending on the fungal treatment" (L261-262).

Line 379: In general I like this paragraph a lot and it brings up many good points, but this statement has me confused because weren't the differences in total soil C with fungal inoculation due to stabilizing the existing C rather than adding new (fungal-derived or plant-derived) C?

Response:

Yes, the differences in total soil C were due to reduction in decomposition of existing C. We agree that the sentence as written can lead to confusion. What is meant in this sentence is that the influence of the treatments on the microbial community (increase in fungi to bacteria ratio) could have contributed to the stabilising of the existing C. We have modified the text accordingly to increase clarity (L418).

Technical corrections:

Line 137: typo (18 weeks)

Figure 2 : In the y axis label add "C" between ug and g

Response:

These have been fixed (L156).

Reviewer 2:

This study aims to determine the effects of non-mycorrhizal root-associated fungi on soil C cycling and to unravel the underlying mechanisms. I like the research topic since most studies focus on the roles of mycorrhizal root-associated fungi in SOC dynamics. A lot of work has been done, and the manuscript is well-written. I think this manuscript is suitable for Biogeoscience and recommend a major revision. Please see below for my comments.

Main concerns:

First, it is important to define the terminologies before using them. For example, what do "stability" and "persistence" mean since different soil scientists may have different ideas? In my opinion, SOC "stability" means the content of MAOC, but obviously, it is not in this study. To avoid the confusion, I suggest authors define the meanings of "stability" and "persistence" at least for this study before using them.

Response:

Thank you for raising this point. We agree that these terms need to be defined and simplified to avoid confusion. We use the term soil C stability/stabilisation to refer to the resistance of soil C to decay, and have included this definition in the text (L37-38). We have replaced use of the term persistence with stability. We have made modifications throughout the text to ensure consistent use of these terms.

Second, inconsistent fungi effects on SOC were found between two methods (modeling and SOC fraction measurement). For example, some fungi treatment increased resistance SOC pool based on the modeling work; however, insignificant effects of fungi on MAOC were found based on SOC fraction measurement. I prefer trusting the "real observation" (the MAOC value) instead of "the estimated value" (the modeled resistance C pool). Some part of the conclusion (e.g. Line 17) is based on the model's outputs. Please see more details of my ideas from the specific comments. At least, the limitation of model should be mentioned in the revised manuscript.

Response:

Here we take the opportunity to clarify that we used various assessments of C responses: total C, plant- versus soil-derived C, mineral associated organic C, aggregated C, and finally, labile, intermediate, and resistant C. These last three fractions are based on the dynamics of C decomposition during long term incubation. The dynamics of decomposition were measured empirically, quantifying $CO_2$ release over time. The behaviour of the $CO_2$ release was fitted to an exponential decay curve and the parameters of this curve were calculated to obtain the sizes of the labile, intermediate and resistant

C pools from which $CO_2$ is released. Thus, these pools are obtained from empirical assessments, not from modelling. We use the term model as a short term to refer to the exponential decay equation. We acknowledge this term may lead to confusion as it could be interpreted as simulation modelling of some kind. We have clarified what we mean by this term in the text (L191).

Third, I suggest authors run SEM analysis to give a big picture of how fungi affect SOC cycling by influencing root, plant, SOC fractions.

Response:

Thank you for the suggestion to perform structural equation modelling to estimate the importance of pathways by which fungi influence soil carbon formation. We did consider this but decided against it because the current study design isn't well suited due to low sample size. Our independent unit of replication here is each fungal isolate, and with only 12 of these in our study we can only confidently analyse very simple models with only 2-3 parameters (Eisenhauer et al., 2015, https://doi.org/10.1016/j.pedobi.2015.03.002).

Specific comments:

Figure A1: It would be better to move the panel named "Plant growth" to the very left. Otherwise, readers may think he first step of the "Wheat plant growth experiment" is "Soil C attributes" instead of "Plant growth".

Response:

This has been fixed.

Line 115: Can you please list the plant species that were used to isolate twelve fungal?

Response:

The fungi were isolated from multiple species (primarily grasses and shrubs) including: *Chloris truncata*, *Paspalum* sp., *Poa sieberiana*, *Austrostipa* sp., and *Enchylaena tomentosa*. We have added to the text that multiple species of grasses and shrubs were used (L122-123).

Line 121: How about the land use history? Is it wheat system?

Response:

The past 10 years of land use history for the soil included wheat, barley, canola, and sorghum. We have added this information to the text (L132-133).

Line 124: I am confusing here. Why and how were the pots distributed among "six" chambers? In addition, how big of each pot (the dimension)? How many soils were contained in each pot?

Response:

The pots were distributed among six climate- and $CO_2$-controlled growth chambers. Each chamber contained one replicate per treatment for replicates 1 to 6, and replicate 7 was distributed among the chambers. The pots were 2 L and contained 1800g soil each. This section has been modified to increase clarity (L135-139).

Line 129: Why were two instead of three agar squares added to "unplanted" control pots to keep the number of agar square the same between treatments?

Response:

Two agar squares were used in the control pots as these pots were smaller and contained less soil (500 g) than the inoculated pots (1800 g). We have added this to the text (L147-149).

Line 130: Can you please give more information on how this number "142" was calculated? I know seven planted replicates inoculated with one of the 12 fungal isolates should be 84 pots plus 6 replicates of uninoculated planted pots, which should be 90.

Response:

In addition to the 84 inoculated planted pots and six uninoculated planted pots, there were also four replicates of "unplanted" pots containing only fungal inoculum for fungal treatment (including no isolate controls), adding to 142 pots in total. We have clarified this in the text (L147-149).

Line 162: A subscript is essential for some abbreviation (e.g. CUP-Soil, CSoil, and CP).

Response:

This has been fixed (L181).

Line 170: What were the standard temperature and moisture?

Response:

The incubations were performed at 25oC, and gravimetric moisture content of the soil was 42%. This information is included in the supplementary information but has been added to the main text (L190).

Line 174: How and how often was CO2 rate measured?

Response:

The following information is included in the supplementary information but more details (such as the number of measurements) have been added to the main text (L189-191):

"Headspace samples (40 mL) were collected on 16 occasions over the course of 135 days (eight times in the first two weeks, and less frequently thereafter). Prior to headspace sampling jars were opened to allow equilibration with ambient air outdoors and then closed. Jars were then immediately placed in the incubator for periods ranging from 24 h during the early days of incubation to 90 h at the final sampling date, to allow approximately 10 000 µmol mol$^{-1}$ $CO_2$ to accumulate. $CO_2$ production rate per hour was calculated based on the length of time after closing. Four jars without soil were used as blanks to account for time zero $CO_2$ concentrations and $\delta^{13}C$ values. Headspace samples were analysed for $CO_2$ concentration with a PICARRO G2201i isotopic $CO_2$/$CH_4$ analyser (Picarro Inc., Santa Clara, California, USA)."

Fig. 2: This figure is very informative. In addition to the absolute value of soil- and plant-derived C, the ratios of soil- and plant-derived C to the total SOC are also important parameters especially for the modeling work. It would be better to calculate these values and do similar analysis like this figure. The relevant results can be put into the supplementary material.

Response:

We have calculated these values and presented them in Fig. A2.

Line 236-237: The p values in Table B2 is different from the values that were mentioned here.

Response:

The p-values mentioned in lines 236-237 are not from the ANOVAs presented in Table B2 but from Pearson's correlation tests comparing soil %C to soil-derived or plant-derived C. We have clarified this in the text (L266).

Table B2: It would be better to run analyses to determine the significance of difference between treatments for each parameter.

Response:

Significant differences between treatments for each parameter were calculated via Dunnett's post-hoc test and are indicated by the asterisks in the table.

Fig. 3-5: Some texts for the variables overlap with each other, making it difficult to recognize them, which should be improved.

Response:

These have now been improved as much as possible.

Line 396-397: The C resistance pool was estimated by model, which was not the direct measurement like MAOC. Therefore, I think we should not conclude this statement because the "real" measurement (MAOC here) was different from the "estimated" value (the resistance C pool here).

Response:

As mentioned above in response to a comment by reviewer 1, decomposition dynamics were measured empirically, quantifying $CO_2$ release over time. The behaviour of the $CO_2$ release was fitted to an exponential decay curve and the parameters of this curve were calculated to obtain the sizes of the labile, intermediate and resistant C pools from which $CO_2$ is released. Thus, these pools are obtained from empirical assessments, not from modelling. We use the term model as a short term to refer to the exponential decay equation. We have clarified this in the text (L191).

Line 399-400: But why does the model show a significant effect of fungi on different SOC pools in a short time (135 days, Fig. 2)? In other words, I wonder the accuracy of the models' predictive ability since we do not see any changes in MAOC and POC under fungal treatment (the real observation). Since the pools in the current SOC models are not measurable (see Lavallee et al. 2019, Global Change Biology), I would trust the SOC fractions data more.

Response:

We clarify that by model we just mean the exponential decay curve that we fitted to the observations of 4 months of $CO_2$ release. This decay curve is not really making predictions but simply calculating the size of the pools that would generate the dynamics of $CO_2$ production that are observed (i.e. real observations). In 4 months we observed loss of C from a very labile pool (steep decline phase) and intermediate pool (the medium slope phase). The resistant pool is calculated by difference. Thus, these are functionally measured pools based on actual, or "natural", processing and retention or loss of carbon from soil after exposure to the experimental treatments. Parameters derived from mid- to long-term soil incubation data are sensitive measures of changes in the distribution and stability of C pools resulting from previous exposure to experimental treatments (Carney et al. 2007, Carrillo et al. 2011, Jian et al. 2020, Langley et al. 2009, Taneva & Gonzalez-Meler 2008).

We acknowledge that the notion of estimating pools and fractions of C in soil is a necessary approach to address soil C complexity and that soil C exists along a continuum of properties. Thus all methodologies are useful but are imperfect and have limitations. For instance, the density and size determined fractions obtained via fractionation protocols are defined by size/density operator-defined thresholds (for example the < 53 microns to determine MAOC) and are considered to be potential indicators of C protection/stability, with the recognition that the 53 microns is only an approximation and large variation exists in nature and thus this size may under or overestimate actual mineral association. Equally, what is considered labile/intermediate and resistant would depend on soil/time/conditions. Thus, acknowledging this complexity, our approach was to utilise a multifaceted approach, including functionally (incubation) and operationally (fractionation) defined pools as well as total and plant/soil-derived C. We found that in our case the functional approach was more sensitive to the impacts of treatments. We discussed potential reasons for this in lines 431-442. An implication of these observations is that we should more often combine multiple approaches.

Associate editor suggestions:

- the authors use the terms stability/stable, resistance/resistant and persistence/persistent. It appears to me that these terms all refer to the same thing, i.e. the decrease in decomposition rate of the soil organic matter pool/fraction, or the increase in the size of a pool with relatively low decomposition rates. In that light, perhaps the authors could use the same term throughout. If I am incorrect, and all three terms mean something different, then please provide an explicit definition for all three terms.

Response:

Thank you for raising this point. We agree that these terms need to be defined and simplified to avoid confusion. We use the term soil C stability to refer to the resistance of soil C to decay, and have included this definition in the text (L37-38). We have replaced use of the term persistence with stability. We also use the term "resistant" to refer to the most stable of the three C pools measured via soil incubations. We have made modifications throughout the text to ensure consistent use of these terms.

- L217 why were N levels higher in the treatments that were inoculated/planted? These results seem counter-intuitive, as these treatments did not receive any additional N compared to their control treatments (or did they? in that case, please mention this as the likely explanation).

Response:

No, there was no additional N in any of the treatments. A potential explanation for this difference is that there may have been less N loss in the inoculated treatments due to less decay of organic matter (and its N) and thus less opportunity for N loss via leaching, or volatilisation.

- L235 "exhibited higher amounts of plant-derived C at a level that was marginal in its non-significance" may confuse some readers. Does this mean "tended to contain higher levels of plant-derived C (add p-value here)"

Response:

This has been changed (L263-264).

- L348-349 "However... soil C storage." sounds a bit convoluted. Perhaps you could simplify it to "However, soil C storage can also be achieved through reductions in soil C outputs."

Response:

This has been changed (L386-387).